# FtsZ treadmilling is essential for Z-ring condensation and septal constriction initiation in *Bacillus subtilis* cell division

Kevin D. Whitley [1,2,5], Calum Jukes[1,5], Nicholas Tregidgo[1], Eleni Karinou [1], Pedro Almada[3], Yann Cesbron [1], Ricardo Henriques [3,4], Cees Dekker [2] & Séamus Holden [1✉]

Despite the central role of division in bacterial physiology, how division proteins work together as a nanoscale machine to divide the cell remains poorly understood. Cell division by cell wall synthesis proteins is guided by the cytoskeleton protein FtsZ, which assembles at mid-cell as a dense Z-ring formed of treadmilling filaments. However, although FtsZ treadmilling is essential for cell division, the function of FtsZ treadmilling remains unclear. Here, we systematically resolve the function of FtsZ treadmilling across each stage of division in the Gram-positive model organism *Bacillus subtilis* using a combination of nanofabrication, advanced microscopy, and microfluidics to measure the division-protein dynamics in live cells with ultrahigh sensitivity. We find that FtsZ treadmilling has two essential functions: mediating condensation of diffuse FtsZ filaments into a dense Z-ring, and initiating constriction by guiding septal cell wall synthesis. After constriction initiation, FtsZ treadmilling has a dispensable function in accelerating septal constriction rate. Our results show that FtsZ treadmilling is critical for assembling and initiating the bacterial cell division machine.

[1] Centre for Bacterial Cell Biology, Biosciences Institute, Faculty of Medical Sciences, Newcastle University, Newcastle upon Tyne NE2 4AX, UK. [2] Department of Bionanoscience, Kavli Institute of Nanoscience, Delft University of Technology, Van Der Maasweg 9, Delft 2629 HZ, The Netherlands. [3] MRC Laboratory for Molecular Cell Biology, University College London, London WC1E 6BT, UK. [4] Instituto Gulbenkian de Ciência, Oeiras 2780-156, Portugal. [5] These authors contributed equally: Kevin D. Whitley, Calum Jukes. ✉email: seamus.holden@ncl.ac.uk

Bacterial cell division is an essential cellular process and a key target for antibiotics. Bacterial cell division is also a remarkable feat of molecular self-assembly where a nanoscale divisome machine builds a micron-scale wall (septum) at mid-cell. The core components of this machine include the septal cell-wall synthases that insert new peptidoglycan (PG) to build the septum, and the essential cytoskeletal tubulin homologue FtsZ that polymerises into a dense Z-ring band of short filaments at mid-cell[1–3]. These short FtsZ filaments move by treadmilling, a type of motion where an asymmetric filament undergoes plus-end polymerisation and minus-end depolymerisation, with GTP hydrolysis setting the rate of FtsZ depolymerisation and overall treadmilling speed[4–6]. FtsZ treadmilling dynamics are essential for cell division[6–9].

However, the role of FtsZ treadmilling in cell division remains unclear. In the Gram-positive model organism *Bacillus subtilis*, we previously observed that constriction rate and synthase speed are dependent upon FtsZ filament speed[5], leading to an initial model of a tightly coupled motile cytoskeleton-synthase complex, where FtsZ acts an obligatory guide and rate-limiting step for septal PG synthesis. In contrast, FtsZ treadmilling is dispensable after constriction initiation in *Staphylococcus aureus*[7], even though the two organisms belong to the same phylogenetic class. This could reflect genuine differences between species, or there may exist a common division mechanism where FtsZ treadmilling plays different roles at different division stages. To date, it remains unclear how the role of treadmilling FtsZ filaments varies across the full division process.

Here, we measure the dynamics and function of FtsZ treadmilling across all division stages in *B. subtilis*. To make these observations, we developed a method combining nanofabrication, advanced microscopy and microfluidics. This approach allowed us to measure native FtsZ dynamics at near-single-filament resolution throughout division. We found that treadmilling dynamics drive FtsZ filament interactions during Z-ring assembly and that this stabilises filaments into dense Z-rings. We determined that the FtsZ inhibitor PC190723 arrests FtsZ treadmilling within seconds, independent of division stage, allowing us to elucidate the rapid response of cells to treadmilling arrest. By systematically determining the response of single cells to FtsZ treadmilling arrest by PC190723, we found that FtsZ treadmilling has two separate essential functions: mediating Z-ring assembly during the earliest stage of cell division, and subsequently for guiding septal constriction initiation. We found that FtsZ treadmilling becomes dispensable after constriction initiation but increases septal constriction rate, reconciling the findings from previous studies[5,7]. Our results show that the key roles of FtsZ treadmilling are to drive Z-ring assembly and to initiate septal constriction.

## Results

**FtsZ assembly at mid-cell has three distinct phases: nascent, mature and constricting Z-rings.** We first determined FtsZ organisation during cell division in *B. subtilis* using an FtsZ-GFP fusion[10] expressed from the native locus at near wild-type protein levels (SH130, Methods, Supplementary Fig. 1). Cell morphology analysis showed near native morphology with only a mild elongation phenotype at both 30 °C and 37 °C (Supplementary Fig. 2), although growth in liquid culture was not impaired at either temperature (Supplementary Fig. 3).

We performed time-lapse imaging of FtsZ-GFP organisation in slow growth conditions (Fig. 1 and Supplementary Video 1). We observed that FtsZ filaments initially formed a diffuse structure at mid-cell which rapidly condensed into a dense narrow band, followed by the onset of constriction (Fig. 1a). We quantified

these changes in Z-ring structure by measuring the dimensions (thicknesses and diameters) and septal density (total septal intensity divided by Z-ring circumference) of the Z-rings in each frame (Methods). The resulting traces of thicknesses and diameters over time were aligned by the start time of constriction (Methods, Fig. 1b). These time traces revealed three distinct Z-ring stages that we classified as 'nascent', 'mature', and 'constricting' based on Z-ring diameter and axial thickness (Fig. 1b, Methods and Supplementary Note 3). During the nascent Z-ring stage, low density FtsZ filaments transiently assemble at mid-cell over a wide region, consistent with previous reports[8,11–13]. FtsZ filaments then rapidly condense into a thin mature Z-ring structure, which remains unconstricted but increases in intensity for some time. Finally, the Z-ring begins constricting until division is finished. Step finding analysis of Z-ring axial thickness time traces showed that Z-ring condensation occurs rapidly and stochastically ~10 min before constriction initiation (Fig. 1b, Methods and Supplementary Fig. 4).

**VerCINI microscopy reveals FtsZ filament dynamics are division stage regulated and FtsZ treadmilling drives filament encounter in nascent Z-rings.** FtsZ filament dynamics are difficult to observe in vivo. TIRF imaging only illuminates a thin slice of the septum, limiting observation to partial snapshots of filament dynamics in nascent Z-rings[5,6]. We previously developed a prototype approach, which resolves this problem by vertically immobilising cells in nanofabricated chambers, allowing continuous imaging of the entire division septum, including dense actively constricting Z-rings[5,8]. However, the prototype method was not capable of imaging dim, nascent Z-rings due to high background, suffered from fast photobleaching and lacked the throughput required.

We created an entirely redesigned vertical cell immobilisation method which addresses all of these issues and allows ultrasensitive high-resolution imaging of FtsZ filament dynamics at all stages of division, termed VerCINI (vertical cell imaging by nanostructured immobilisation). This method features a high-throughput chip design, centrifugation loading, high SNR ring-HiLO illumination[14], imaging denoising[15] and custom septal localisation and background subtraction algorithms (Methods, Supplementary Figs. 5, 6 and Supplementary Videos 2, 3). We measured FtsZ filament dynamics during each division stage using VerCINI (Fig. 1e–h, Supplementary Fig. 7 and Supplementary Videos 4–6). Division stage was determined based on septal intensity and diameter (Methods, Supplementary Fig. 8 and Supplementary Note 3).

In nascent Z-rings, we observed that FtsZ filaments are sparsely distributed, with a mixture of treadmilling and immobile filaments (Fig. 1f and Supplementary Fig. 9a, 35% immobile, filament speed <10 nm/s), leading to a low average filament speed with large spread (median 19 nm/s, interquartile range, IQR, 5–31, N = 526). Surprisingly, we frequently observed two or more treadmilling FtsZ filaments colliding, often leading to aggregation and temporary arrest of both filaments (Fig. 1eii and Supplementary Fig. 7). FtsZ filament aggregation by lateral filament interactions is required to bundle FtsZ filaments and condense the Z-ring[16–20]. These data show that FtsZ treadmilling can drive filament aggregation by promoting the filament encounters required for lateral interactions.

Z-ring condensation led to a drop in the number of immobile FtsZ filaments in mature/early constricting Z-rings, yielding a significant increase in average filament speed with narrower speed distribution (15% immobile, median 27 nm/s, IQR 19–34, N = 1053; Fig. 1h). Simulations showed that changes in filament density had little effect on observed filaments speeds or observed

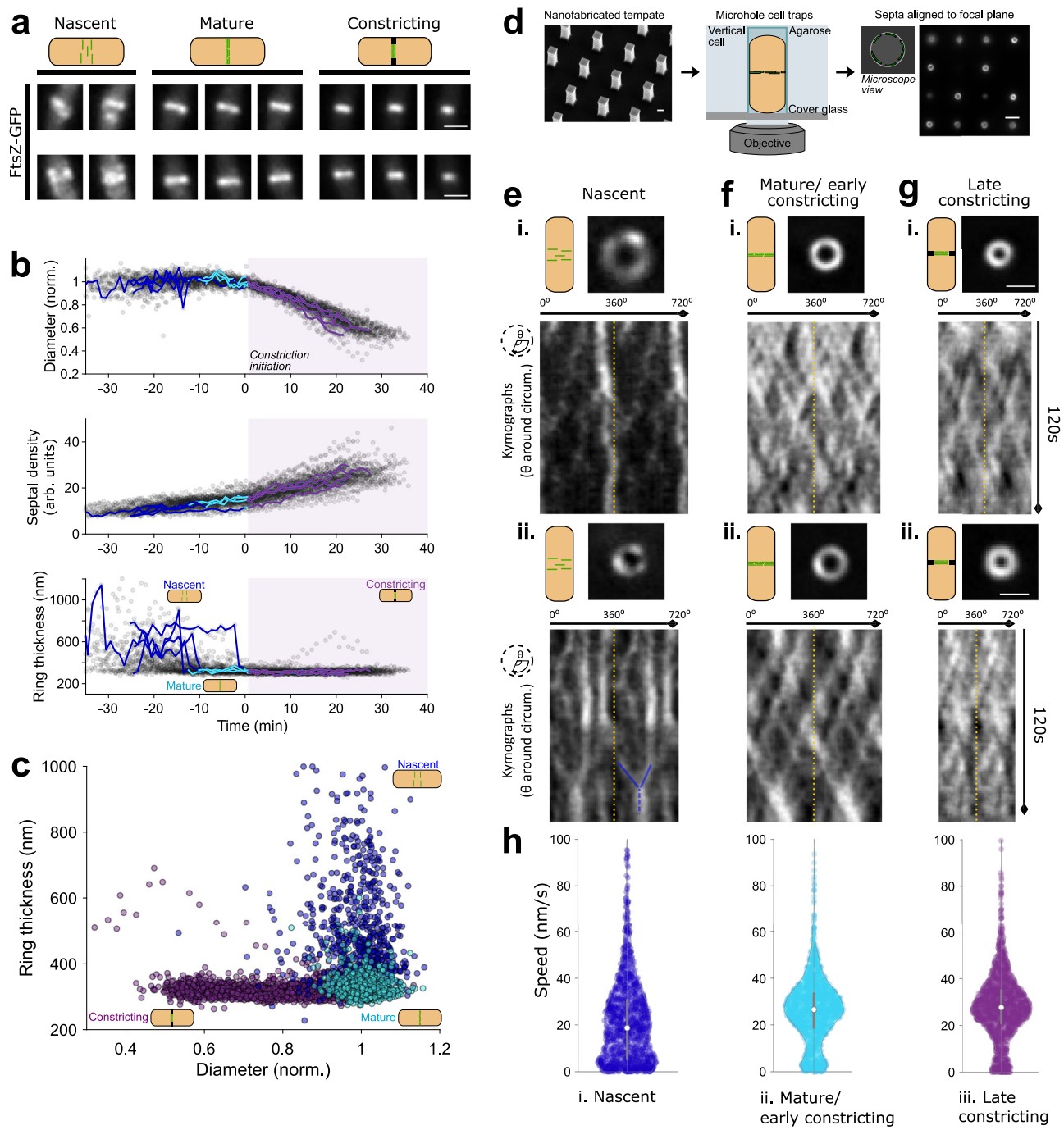

fraction of immobile FtsZ filaments (Supplementary Fig. 10 and Supplementary Note 4). Filament speeds in late constricting Z-rings were similar to those of mature/early constricting Z-rings (13% immobile, median speed 28 nm/s, IQR 21–36, N = 1677; Fig. 1h). Due to the decreased immobile population, filaments in mature and constricting Z-rings traversed, on average, twice the distance around the septum of those in nascent rings (Supplementary Fig. 9c). Furthermore, we found that treadmilling and immobile filaments displayed similar lifetimes (Supplementary Fig. 11), suggesting that immobile filaments are dynamic. These results show that the fraction of treadmilling FtsZ filaments and their speeds are division stage regulated.

**PC190723 arrests FtsZ treadmilling within seconds throughout division**. We next set out to determine the function of FtsZ

treadmilling during each stage of cell division. For this we needed to observe the effect on cell division in response to rapid FtsZ treadmilling arrest. The antibiotic PC190723 (PC19) arrests FtsZ treadmilling in *B. subtilis* and related species by supporting the 'open' conformation of FtsZ and stabilising protofilaments[5,21–24] without fully abolishing subunit turnover or GTPase activity[25]. Since the entire cell division process takes only 10–20 min in fast growth conditions, it was critical to determine whether PC19 acted sufficiently rapidly and robustly for use in dissecting the effect of treadmilling during each division stage. To determine how PC19 and other cell division inhibitors perturb FtsZ tread-milling, we developed microfluidic VerCINI, which combines high-resolution imaging of cell division protein dynamics in vivo with rapid chemical perturbation (Methods, Fig. 2a). In this method, cells are confined in an array of open-topped microholes

**Fig. 1 FtsZ filament organisation and dynamics throughout the division process. a** Exemplar images of FtsZ-GFP (SH130) filament organisation throughout division, classified by division phase. Scale bars: 1 μm. All images same magnification. **b** Quantification of FtsZ-ring diameter, septal density (septal intensity divided by ring circumference) and ring thickness throughout division from time-lapse microscopy data. Nascent Z-rings have large axial width (ring thickness) due to the diffuse distribution of filaments, which condense into a thin mature Z-ring, followed by constriction initiation. Traces are temporally aligned relative to the start time of constriction (Methods). Grey scatter points represent all data points. Lines show individual, representative traces, which are split into coloured segments indicating cell division state: nascent (blue), mature (cyan) and constricting (purple). Purple shading shows all time after constriction initiation. **c** FtsZ-ring thickness and relative septal diameter. Colour indicates cell division state as per (**b**). Cell division state of FtsZ-rings in (**b**, **c**) determined by automated classification of Z-ring diameter and axial thickness time lapse data (Methods, Supplementary Note 3). **d** VerCINI schematic. Nanofabricated silicon micropillars (left panel) are used as a mould to make agarose microholes. Rod-shaped bacteria are trapped in agarose microholes (middle panel), rotating the division septum into the microscope imaging plane (right panel). Scale bars: 1 μm (left panel), 2.5 μm (right panel). **e–g** VerCINI microscopy of FtsZ filament dynamics for each Z-ring phase, two representative examples per phase (i–ii). Nascent Z-rings are composed of sparse filaments diffusely distributed around the circumference of the cell, and a large fraction of immobile FtsZ filaments. Mature and constricting Z-rings possess a more uniform distribution of FtsZ filaments around the division site, with most filaments treadmilling. Images of septa show the first frame in kymographs (t = 0 s). Kymographs were obtained by fitting septal images to circles and plotting intensity values around the circumference of the cell for each frame of the time-lapse (1 frame/s). Two full revolutions around the cell (0–720º) are plotted side-by-side in each kymograph to resolve filament trajectories that pass 0º/360º, separated by yellow dotted lines. Blue lines highlight example of motile FtsZ filament aggregation. Scale bars: 1 μm. All images same magnification. **h** Violin plots of FtsZ filament speed for each division stage. White circles, median; thick grey lines, interquartile range; thin grey lines, 1.5x interquartile range. Source data for this figure are provided as a source data file.

formed from polydimethylsiloxane (PDMS) atop a microscope coverslip, which is contained within a microfluidic chamber to allow continual flow of fresh media as well as chemical inhibitors. PDMS was used rather than agarose in order to produce a thin enough layer (~50 μm) to image through. Cells continued to grow and divide, consistent with previous observations that growth of *B. subtilis* cells in sufficiently wide (>0.8 μm) PDMS channels is not impaired[26], and models demonstrating that nutrient transport does not significantly limit growth in short (<20 μm) channels[27]. A real-time image-based autofocus system was used to stabilise focus during high-speed imaging[28].

We imaged FtsZ-GFP (SH130) using microfluidic VerCINI to observe the effects of division inhibitors PC19 and Penicillin G (PenG) on FtsZ treadmilling. We recorded untreated FtsZ dynamics for 90 s while blank media flowed over the cells, then continuously treated cells with an excess of division inhibitor by changing fluid flow to media laced with the compound (Fig. 2b and Supplementary Video 7). FtsZ treadmilling dynamics were abolished by PC19 within a few seconds for all diameters of septa observed (Fig. 2c). When we treated cells with PenG, which inhibits cell wall synthesis by binding to cell wall synthase transpeptidation sites, FtsZ dynamics continued unperturbed for the duration of the experiment (several minutes). FtsZ dynamics were also unaffected by a DMSO-only control, the solvent for PC19 (Fig. 2c and Supplementary Videos 8, 9).

By providing high spatiotemporal resolution during rapid chemical perturbation, microfluidic VerCINI revealed the effect of division inhibitors on FtsZ treadmilling throughout division. Our results demonstrate that PC19 inhibits FtsZ treadmilling within seconds well after the start of constriction, whereas PenG treatment does not directly affect FtsZ dynamics.

**FtsZ treadmilling is essential for Z-ring condensation and septal constriction initiation, but is subsequently dispensable.** We used PC19 to investigate the role of FtsZ treadmilling throughout cell division. We imaged cells in fast growth conditions (rich media, 37 °C) using a commercial microfluidic device with continual fluid flow both before and during treatment with excess PC19 (Fig. 3, Methods). Treatment with PC19 prevented Z-rings in earlier stages of division from continuing to divide, but Z-rings in later stages continued dividing (Fig. 3a, b and Supplementary Videos 10–13). PC19 treatment prevented 96% (N = 62) of nascent Z-rings from condensing into mature Z-rings (Supplementary Fig. 13) and 100% (N = 66) from constricting (Fig. 3b). Furthermore, we found that treatment prevented 71%

(N = 56) of mature Z-rings from beginning constriction, while only 5% (N = 66) of initially constricting Z-rings were prevented from continuing constriction. In contrast, treatment with PenG inhibited constriction altogether (Supplementary Video 14). Similar results were observed in slow growth conditions (poor media, 30 °C; Supplementary Fig. 14 and Supplementary Video 15). Together these data suggest that FtsZ treadmilling is required for the condensation of filaments from a nascent to a mature Z-ring, but becomes dispensable at some point during the mature Z-ring stage.

It is worth noting that PC19 also affects FtsZ GTPase activity and filament structure[21]. These are likely downstream effects of the primary mode of action, which is to stabilise the polymeric state of the protein. We note that FtsZ retains ~25% GTPase activity even with excess PC19[21]. Furthermore, in cells treated with 8j (a relative of PC19) the subunits in immobile FtsZ foci still turn over slowly[25], which suggests filaments are still dynamic despite not treadmilling. So, while we cannot strictly rule out the possibility that our observations result from a downstream effect of PC19, we believe the most parsimonious explanation for our results is that the arrest of treadmilling specifically is responsible.

We considered whether these effects were specific to PC19 by repeating the assay with the non-benzamide FtsZ inhibitor PC58538 (PC58). This compound inhibits FtsZ by a separate mode of action from PC19: Rather than stabilising protofilaments, it prevents filament assembly[29]. As with PC19, PC58 prevented 84% (N = 50) of nascent Z-rings from condensing into mature Z-rings and 90% (N = 50) from constricting, but prevented 60% (N = 52) of mature and only 6% (N = 33) of constricting Z-rings from constricting (Supplementary Fig. 15 and Supplementary Video 16). Interestingly, similar results were recently demonstrated using ADEP antibiotics, which activate the protease ClpP to degrade FtsZ monomers and thereby deplete the cytoplasmic pool needed for filament assembly[30]. We also repeated the above experiment with exogenous addition of *B. subtilis* MciZ, although we found that the timescale of action was too slow to draw conclusions about its effect on Z-ring condensation or constriction (Supplementary Video 17). We conclude that arresting FtsZ treadmilling with PC19 produces an effect similar to, and as severe as, preventing FtsZ filament assembly altogether, emphasising the critical role of FtsZ treadmilling in cell division initiation.

To further investigate the role of FtsZ treadmilling in Z-ring condensation, we examined the effect of an FtsZ mutant that is deficient in GTP hydrolysis. The mutant FtsZ(D212A) in *E. coli*

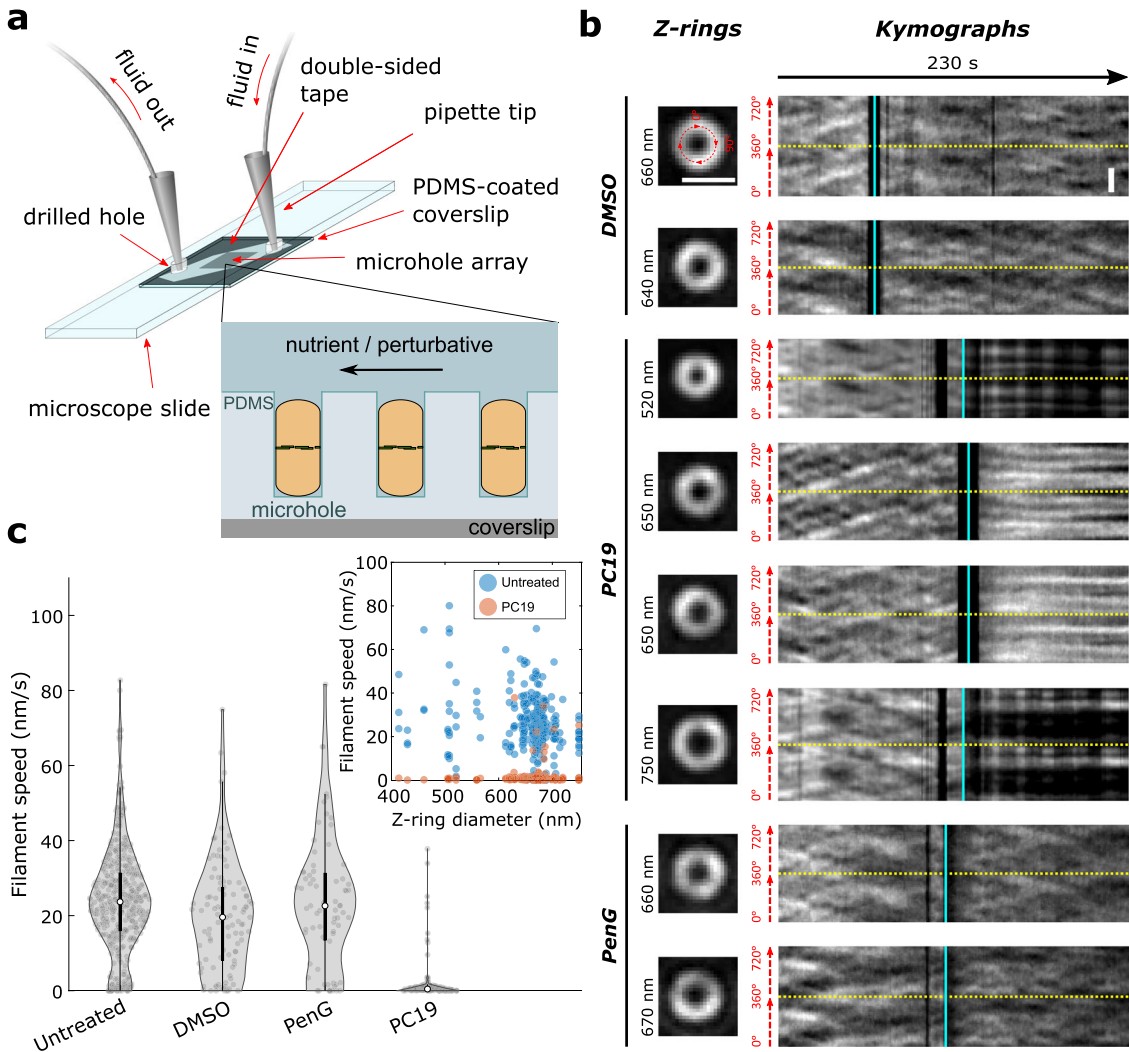

**Fig. 2 PC190723 arrests cellular FtsZ treadmilling within seconds across all stages of constriction. a** Schematic of microfluidic VerCINI. Rod-shaped cells are confined in open-topped microholes in a thin layer of PDMS atop a microscope coverslip as solutions flow over them. The flow channel is formed from a cut piece of double-sided tape sandwiched between the PDMS layer and a microscope cover slide with drilled holes to allow inlet and outlet tubes. **b** Representative images of septa and associated kymographs of FtsZ-GFP (SH130) dynamics in vertically-immobilised cells during rapid treatment with either DMSO (top), PC19 (middle) or PenG (bottom) at multiple septal diameters. Images of septa show the first frame in kymographs ($t = 0$ s). Kymographs were obtained by fitting septal images to circles and plotting intensity values around the circumference of the cell for each frame of the time-lapse (1 frame/s). Two full revolutions around the cell (0–720°) are plotted side-by-side in each kymograph to resolve filament trajectories that pass 0°/ 360°, separated by yellow dotted lines. Cyan lines show arrival time of media containing DMSO, PC19 or PenG. Black bands around time of treatment resulted from a loss of focus during fluid exchange. Fluctuations in intensity for two septa post-PC19 treatment (750 and 520 nm diameter septa) resulted from manual refocusing during imaging. Scale bars: 1000 nm. All images same magnification. **c** Violin plots of FtsZ treadmilling speeds pre- and post-treatment measured from kymographs. White circles, median; thick black lines, interquartile range; thin black lines, 1.5x interquartile range. DABEST plots of effect size can be found in Supplementary Fig. 12. Inset: FtsZ treadmilling speed distributions for untreated (blue circles) and PC19-treated (red circles) cells from violin plots separated by septal diameter. Source data for this figure are provided as a source data file.

was shown to have greatly reduced GTPase activity[31,32], and exogenous expression of the equivalent FtsZ(D213A) mutant in *B. subtilis* reduces FtsZ treadmilling speed[5] (Supplementary Fig. 16). We constructed a merodiploid strain expressing FtsZ-GFP from the native locus and FtsZ(D213A) from a secondary locus under an IPTG-inducible promoter (SH131, Methods). For comparison to wild-type (Fig. 1) we imaged this strain under slow growth conditions (poor media, 30 °C) with an induction level (10 μM IPTG) that supported growth in liquid culture (Supplementary Fig. 3) and cell division (Supplementary Video 18). Condensation of nascent Z-rings was significantly perturbed under these conditions: 54% ($N = 74$) of nascent Z-rings failed to fully condense into narrow Z-rings prior to constriction

initiation, and even throughout constriction Z-rings remained 65 nm (95% CI [63, 67]) thicker than wild-type (Supplementary Fig. 17), consistent with previous measurements in *E. coli*[33]. This partial Z-ring condensation phenotype is consistent with partial inhibition of FtsZ treadmilling caused by expression of the GTPase-deficient mutant.

We investigated how FtsZ(D213A)-expressing cells could still constrict despite forming thick partially condensed Z-rings. First, we performed state detection on the D213A Z-rings to attempt to identify transitions between a nascent and condensed state. However, unlike in wild-type cells, we could only reliably detect a single pre-constricted Z-ring state in the D213A cells (Supplementary Fig. 18a, b). We thus compared the axial thickness of the

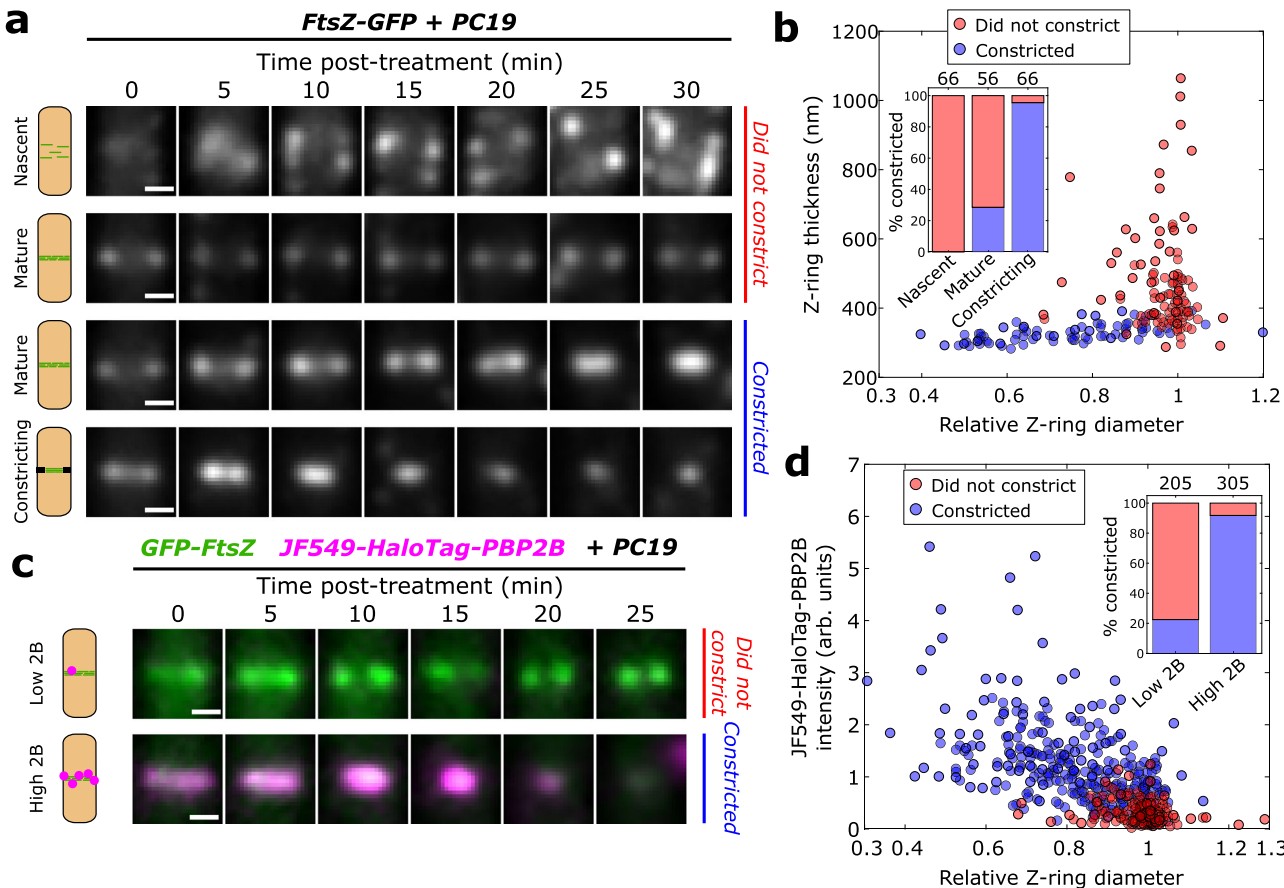

**Fig. 3 FtsZ treadmilling is required until the arrival of PG synthesis machinery. a** Representative time-lapses of Z-rings for FtsZ-GFP cells (SH130) after arrival of 10 μM PC19-laced media. Nascent rings and many mature rings do not constrict for tens of minutes after PC19 treatment, whereas constricting rings and many mature rings continue constricting after treatment. **b** Scatter plot of Z-ring diameters and thicknesses for all FtsZ-GFP cells at $t = 0$ min showing whether they continued constricting (blue) or not (red). Inset: Percentage of cells that continued constricting classified by division stage. Nascent: thickness >400 nm, mature: thickness <400 nm, relative diameter >0.9, constricting: thickness <400 nm, relative diameter <0.9. **c** Representative time-lapses of rings from two-colour strain SH212 (Green: GFP-FtsZ, magenta: JF549-HaloTag-PBP2B) after the arrival of PC19-laced media. Z-rings with low JF549-HaloTag-PBP2B signal typically do not continue constricting while those with high signal typically do. **d** Scatter plot of Z-ring diameters and normalised JF549-HaloTag-PBP2B intensities at $t = 0$ min showing whether they continued constricting (blue) or not (red). Inset: Percentage of cells that continued constricting classified by JF549-HaloTag-PBP2B signal. Low 2B: intensity <0.5, High 2B: intensity >0.5. Scale bars: 500 nm. Numbers above stacked bars indicate number of cells. Source data for this figure are provided as a source data file.

D213A Z-rings to that of wild-type nascent and mature Z-rings. We found that the thickness of the D213A Z-rings is intermediate between that of wild-type nascent and mature Z-rings. We also noticed that most pre-constricted D213A Z-rings looked quite dissimilar to the highly dynamic nascent Z-rings seen in wild-type cells, but instead appeared similar to the less dynamic mature Z-rings except with aberrant shapes (Supplementary Videos 1 and 18). This prompted us to measure the variation in Z-ring axial thickness over time for both wild-type and D213A-expressing cells. We found that the variations in thickness for pre-constricted mutant Z-ring structures were small and comparable to those of mature wild-type Z-rings (Supplementary Fig. 18d). On the other hand, the axial thickness of nascent Z-rings in wild-type cells was highly variable, consistent with their visibly dynamic appearance, and was considerably more variable than either wild-type mature Z-rings or FtsZ(D213A) pre-constricted Z-rings (Supplementary Fig. 18c). Taken together these results show that in cells expressing GTPase-deficient FtsZ (D213A), FtsZ filaments rapidly aggregate into partially condensed Z-rings which are dense and structurally stable, but which have increased axial thickness compared to mature Z-rings in wild-type cells.

To further investigate the connection between condensation and septal constriction, we imaged wild-type and D213A-expressing cells with a membrane stain (Supplementary Fig. 18e). In the mutant, we observed sharp membrane invaginations in many cases where aberrant Z-ring shapes occurred, indicating active but aberrant constriction. In contrast, we did not observe any clear cases of membrane invagination over areas corresponding to nascent Z-rings in wild-type cells. These observations indicate that the rapid treadmilling dynamics of FtsZ filaments in wild-type nascent Z-rings are likely required to form a compact Z-ring structure to mark the division septum to enable constriction. Reduced filament treadmilling and turnover in the FtsZ(D213A) mutant still allow assembly of a dense Z-ring structure, but locked into a thick, partially-condensed state that is less effective in directing cell division. This observation is reminiscent of how dynamic instability of microtubules allows them to rapidly conform to the complex intracellular geometry of eukaryotic cells.

We next wondered what events within the mature Z-ring stage corresponded to the change in dispensability of FtsZ treadmilling. Divisome assembly in *B. subtilis* is approximately a two-step process where FtsZ and other cytoplasmic proteins arrive first,

followed by arrival of the PG synthesis machinery just before constriction initiation, including the septal transpeptidase PBP2B[34]. To determine more precisely when FtsZ treadmilling becomes dispensable, we repeated the treadmilling inhibition assay using a two-colour strain where FtsZ and PBP2B are both labelled. Since fully-labelled FtsZ slightly perturbed cell physiology (Supplementary Fig. 2) and full labelling of FtsZ was unnecessary for this assay, for the labelled FtsZ we decided to use a well-characterised merodiploid harbouring an N-terminal GFP-FtsZ fusion[25] that could be expressed at a low non-perturbative level. We therefore constructed a two-colour strain (SH212, Methods) expressing GFP-FtsZ from an ectopic locus and HaloTag-PBP2B from its native locus. To quantify the amount of PBP2B at septa, we labelled HaloTag-PBP2B with an excess of JF549 HaloTag ligand[35] (Methods). This strain showed the same response overall to PC19 treatment as before (Supplementary Fig. 19 and Supplementary Videos 19, 20). We found that the amount of PBP2B present at time of FtsZ treadmilling arrest correlated strongly with whether rings would constrict or not (Fig. 3c, d). Similar results were obtained for a functional mNeonGreen-PBP2B fusion expressed from the native locus[5] (ME7, Supplementary Video 21 and Supplementary Fig. 20). The GFP-FtsZ signal also increased, but unlike PBP2B the FtsZ signal was only weakly correlated with the percentage of cells continuing constriction (Supplementary Fig. 19b). FtsZ treadmilling thus becomes dispensable for cell division just after the arrival of the PG synthesis machinery. These results are consistent with previous findings in *S. aureus* that FtsZ treadmilling is required until constriction has initiated, and thereafter becomes dispensable[7]. Since we previously observed that FtsZ treadmilling guides processive PG synthesis around the septum[5], we conclude that FtsZ treadmilling in mature Z-rings likely functions by guiding the processive motion of septal synthases during the earliest stages of septal synthesis, before a septum has been formed.

Together, our data reveal two essential functions for FtsZ treadmilling in division. The first is a PG synthesis-independent function, to condense sparse FtsZ filaments into a dense Z-ring. The second function is to initiate constriction by guiding initial cell-wall synthesis.

**FtsZ treadmilling accelerates constriction**. We wondered how to reconcile the finding that FtsZ treadmilling is dispensable for constriction after initiation with our previous observation that FtsZ treadmilling sets the rate of constriction[5]. To address this, we set out to determine the effect of rapid, total arrest of FtsZ treadmilling on the septal constriction rate using natively-expressed mNeonGreen-PBP2B as a septal marker (ME7). In order to precisely measure constriction rates immediately after PC19 treatment, we developed a physical model for Gram-positive septal constriction which could be fitted to partial septal constriction trajectories immediately after cell treatment with PC19 (Methods, Supplementary Note 2 and Fig. 4a). For comparison to previous observations we converted the observed constriction rates of partially constricted cells to an effective constriction time $t_{eff}$, the total amount of time it would take for an average septum to constrict completely (Methods).

We observed that the constriction time of untreated cells in fast growth conditions was 15.3 min, IQR 13.5–17.9 ($N = 765$, Fig. 4b). FtsZ treadmilling inhibition by PC19 treatment caused a robust 7.7 min, 95% CI [6.9, 8.3] ($N = 419$), increase in constriction time for cells grown in rich media. This is similar to the increase in constriction time caused by high expression of the GTP-hydrolysis-impaired FtsZ(D213A) mutant (Supplementary Fig. 21). Fluorescent PBP2B signal continued to increase after

treatment with PC19 (Supplementary Fig. 22), indicating that the reduced constriction rate is unlikely to be caused by indirect reduction of synthase levels at the septum. These results show that although FtsZ treadmilling is not required for septal PG synthesis after constriction initiation, treadmilling promotes efficient constriction by increasing septal constriction rate. This reconciles apparently contradictory models previously proposed for the role of FtsZ treadmilling in septal constriction[5,7].

**Cell growth rate limits septal constriction rate independently of FtsZ treadmilling**. Growth rate and cell size for most bacterial species are linked to nutrient availability and culture temperature[36]. We sought to identify how growth rate affects the constriction rate of *B. subtilis* cells. We measured the constriction rate in media and temperature conditions (Supplementary Fig. 3c and Supplementary Table 1) using a strain expressing PBP2B from its native promoter fused to mNeonGreen (ME7, Methods). We observed a linear dependence of septal constriction rate on overall cell growth rate (Supplementary Fig. 3d). Slower growth in either minimal media or at low temperature led to correspondingly longer constriction times (Fig. 4b and Supplementary Fig. 3d), with a 2.18-fold (95% CI [2.12, 2.24]) difference in constriction time between fastest and slowest growth conditions.

We considered that this difference in constriction time could result from a difference in the treadmilling speed of FtsZ across growth conditions. We therefore determined FtsZ treadmilling speed using TIRF illumination for all conditions using a dilute exogenous label of mNeonGreen-FtsZ (bWM4) which shows no growth or morphology phenotype at 37 °C[5] (Supplementary Fig. 23). FtsZ treadmilling speed was constant across all growth conditions tested (Fig. 4d), showing that the effect of growth rate on septal constriction is independent of FtsZ treadmilling.

We next investigated whether FtsZ treadmilling also promotes constriction in slow growth conditions, or if cell growth rate becomes the sole rate-limiting factor. As in fast growth conditions, arresting FtsZ treadmilling increased the constriction time (Fig. 4b and Supplementary Video 22). However, the relative increase in constriction time was much lower in slow growth conditions (1.16-fold 95% CI [1.09, 1.24] vs 1.50-fold 95% CI [1.46, 1.55] change). These observations show that cell growth rate is a major regulator of active septal constriction, with treadmilling FtsZ filaments acting as a secondary promotor of septal synthesis.

## Discussion

Using bespoke ultra-sensitive microscopy of cell division protein dynamics, we discovered that FtsZ treadmilling plays two separate essential roles in cell division: to establish the mature divisome by condensing diffuse filaments into a dense ring, and to guide septal constriction initiation (Fig. 5).

We found that condensation of sparse FtsZ filaments into a stable, dense Z-ring structure is required for cell division. Using VerCINI microscopy we observed that treadmilling FtsZ filaments frequently collide and aggregate in nascent Z-rings. We also found that nascent Z-rings contain a large fraction of immobile FtsZ filaments, which do not treadmill, and that Z-ring condensation promotes formation of treadmilling FtsZ filaments. While the mechanism of FtsZ filament motility regulation requires further study, in vitro measurements of FtsZ dynamics suggest that immobile FtsZ filaments could result from low septal FtsZ concentration or regulation by accessory/anchor proteins[4].

Chemical arrest of FtsZ treadmilling completely abolished FtsZ ring condensation and septal constriction. Attractive lateral interactions between FtsZ filaments, via the FtsZ C-terminal-linker and cytoplasmic FtsZ-associated proteins are known to

**Fig. 4 Constriction rate is accelerated by FtsZ treadmilling and fast cell growth rate. a** Time traces of ring diameters from mNeonGreen-PBP2B strain (ME7) for untreated (left) and PC19-treated (right) cells with constant PG synthesis model (red) fitted to representative traces (black). Untreated cell trajectories are aligned relative to fitted constriction start time (black dotted line on left panel). PC19-treated cell trajectories are aligned relative to the arrival time of PC19-laced media (black dotted line on right panel). **b** Constriction time before and after PC19 treatment for both fast growth (rich media 37 °C) and slow growth (poor media, 30 °C) conditions. **c** Constriction time of mNeonGreen-PBP2B cells in rich and poor media at two growth temperatures. **d** mNeonGreen-FtsZ (bWM4) treadmilling speeds in different media and temperature. Top panels, **b**–**d**: violin plots; white circle, sample median; thick black lines, interquartile range; thin black lines, 1.5x interquartile range. Bottom panels, **b**–**d**: Bottom panels: DABEST plots showing effect size analysis, compared to leftmost condition; black circles, median difference; error bars, 95% confidence interval of median difference. Source data for this figure are provided as a source data file.

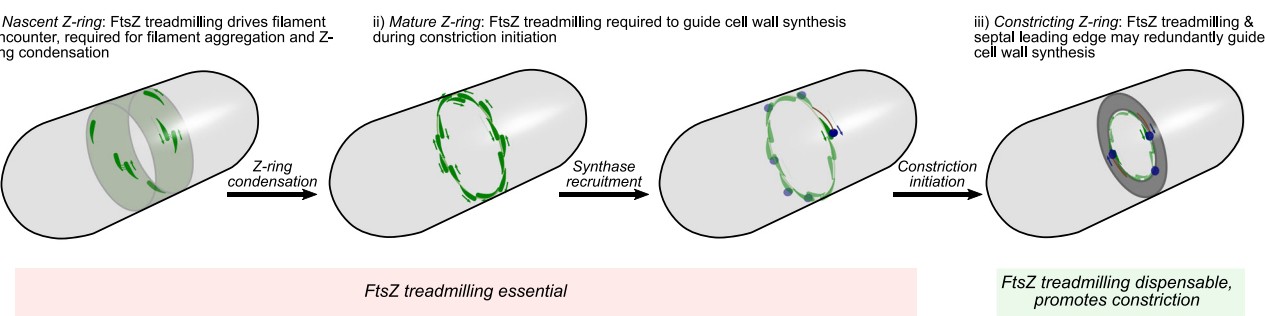

**Fig. 5 Model for function of FtsZ treadmilling in Z-ring assembly and septal constriction.** Schematics depicting stages of division along with key events and roles of FtsZ treadmilling.

mediate FtsZ bundling and Z-ring condensation[16–20]. Our results reveal an additional key factor in assembling the Z-ring: FtsZ filament treadmilling drives filament interactions by enabling FtsZ filaments to rapidly search the mid-cell surface circumferentially and efficiently encounter one another. We additionally found that rapid FtsZ filament turnover is required for full condensation: slow turnover from reduced FtsZ GTPase activity led

to partial filament condensation into aberrantly-shaped Z-rings, and full inhibition of turnover from chemical arrest prevented condensation entirely. These findings contrast with previous models of Z-ring condensation involving lateral sliding of filaments[11,12]. We conclude that FtsZ treadmilling—likely along with FtsZ lateral interactions—drives aggregation of FtsZ filaments into a condensed Z-ring. This is reminiscent of

treadmilling-driven partitioning of FtsZ filaments into higher-order structures previously observed in vitro[4,37].

The second role of FtsZ treadmilling is to guide septal PG synthesis. We found that this role is essential in the constriction initiation phase of *B. subtilis* cell division, but dispensable afterwards. Together with our previous observations that FtsZ treadmilling guides individual synthase molecules during PG synthesis[5], it is likely that during constriction initiation FtsZ filament treadmilling is required to guide the synthases around the circumference of the mid-cell in the absence of an established septum. We also observed that FtsZ treadmilling dynamics are division stage regulated, with Z-ring condensation promoting efficient FtsZ treadmilling. In addition to concentrating the division machinery into a small area, Z-ring condensation may indirectly promote efficient constriction initiation by increasing the fraction of treadmilling filaments available to guide cell-wall synthesis.

Although FtsZ treadmilling is dispensable after constriction initiation, we found that treadmilling increases the rate of septal constriction by up to 1.5-fold. However, we found that overall cell growth rate has a stronger effect on septal constriction rate than FtsZ treadmilling, increasing constriction rate up to 2.2-fold between fast and slow growth conditions. We conclude that FtsZ treadmilling plays a secondary role in active constriction, while its critical roles occur prior to constriction initiation. This contrasts with the previous model of *B. subtilis* FtsZ as an obligatory guide and rate-determining factor for constriction[5]. After constriction initiation, the highly ordered septal leading edge[38] may be sufficient to act as a guide for processive septal PG synthesis, with FtsZ filaments acting as a mid-cell localiser and redundant guide during active constriction.

Surprisingly, we found that the *B. subtilis* septal constriction and overall cell growth rate are strongly coupled, with a linear dependence of septal constriction rate on cell growth rate. The molecular mechanism of coupling between septal constriction and cell growth rate requires further research, but possibilities include control of septal synthesis via overall levels of cell wall precursor[39,40] or levels of cell wall synthesis proteins, or direct regulation of cell wall synthesis protein activity via signalling[41].

The VerCINI method presented here is a powerful tool for ultra-sensitive measurement of cell division protein dynamics and mode of action of cell division targeting antibiotics. We anticipate that many other bacterial structures, especially the bacterial cell envelope and associated proteins, could also benefit from VerCINI's high resolution top-down cell view.

A key puzzle in bacterial cell biology is how nanoscale division proteins spontaneously and accurately build a micron-size wall at mid-cell. Treadmilling gives FtsZ filaments the remarkable ability to autonomously drive their own circumferential motion around the mid-cell surface. Our results show that the self-driving capability of FtsZ filaments is crucial to solving the scale problem of cell division by directing divisome assembly and initial septal synthesis to a narrow band around the mid-cell, laying the template for septum construction. Dynamic, self-organising FtsZ filaments thus provide long-distance order and collective motion around the bacterial mid-cell in the absence of cytoskeletal motor proteins.

## Methods
**Bacterial strains and growth conditions**. Strains used in this study are listed in Supplementary Table 2. Strains were streaked from −80 °C freezer glycerol stocks onto nutrient agar (NA) plates containing the relevant antibiotics and/or inducers and grown overnight at 37 °C. Single colonies were transferred to liquid starter cultures in either time-lapse medium (TLM[42]) or PHMM media and grown with agitation at 200 rpm overnight at either 30 °C (TLM) or 22 °C (PHMM[5]). The next day, TLM starter cultures were diluted to a starting OD600 of 0.1 in chemically-defined medium (CDM[42]), while PHMM starter cultures were diluted to a starting

OD600 of 0.05 in PHMM, and these liquid cultures were grown at 30 or 37 °C with any required inducer until they reached the appropriate OD600. When necessary, antibiotics and inducers were used at the following final concentrations: chloramphenicol 5 μg/mL, spectinomycin 60 μg/mL, erythromycin 1 μg/mL, lincomycin 10 μg/mL, kanamycin 5 μg/mL, xylose 0.08% and IPTG 20 μM.

**Strain construction**. SH211 (PY79 Δ*hag*) was constructed by transforming competent PY79 with genomic DNA extracted from strain PB5250, using standard protocols[43]. The antibiotic cassette in the resulting strain was flanked by Cre recombinase recognition sites (lox71 and lox66), and was subsequently removed as described previously[44]. Briefly, cells were transformed with the temperature-sensitive plasmid pDR244, which expresses the Cre recombinase constitutively, and plated on LB agar supplemented with spectinomycin at 30 °C. On the following day, colonies were re-streaked on un-supplemented LB agar and grown to 42 °C. Strains carrying the markerless version of Δ*hag* were identified by replica plating on un-supplemented LB agar, LB agar + spectinomycin and LB agar + kanamycin. Primers oCJ300 and oCJ301 (Supplementary Table 5) were used to confirm deletion of *hag*.

SH130 (PY79 Δ*hag ftsZ::ftsZ-gfp-cam*) was constructed by transforming SH211 competent cells with genomic DNA extracted from strain PL642. The transformation was verified with PCR. Primer pairs oCJ94/oCJ95 and oCJ94/oCJ06 (Supplementary Table 5) were used to confirm insertion of *ftsz-gfp-cam*.

SH131 (PY79 Δ*hag ftsZ::ftsZ-gfp-cam* Ω *amyE::erm*-P_hyperspank-*ftsAZ*(D213A)) was constructed by transforming competent SH130 cells with genomic DNA extracted from strain bAB215.

SH132 (PY79 *pbp2B::mNeonGreen-15aa-pbp2B* Ω *amyE::erm*-P_hyperspank-*ftsAZ*(D213A)) was constructed by transforming competent ME7 cells with genomic DNA extracted from strain bAB215.

SH212 (PY79 *pbp2B::erm*-P_hyperspank-*HaloTag-15aa-pbp2B* Ω *amyE::spc*-P_xyl-*gfp-ftsZ*) was constructed by transforming competent bGS28 cells with genomic DNA extracted from strain 2020. All published strains are available on request to the authors.

**Bacterial strain characterisation**. Strains were characterised by growth in liquid culture (Supplementary Fig. 3) and cell morphology analysis (Supplementary Fig. 2). Fluorescent fusion protein levels for relevant strains were determined by Western blotting (Supplementary Fig. 1).

**Growth curves**. *B. subtilis* PY79 and variant strains were grown overnight in TLM or PHMM containing the relevant inducers. Overnight cells were diluted to an OD600 of 0.05 in either CDM or PHMM, respectively, and 200 μL were used in a 96-well microtiter plate. Growth was monitored for 12 h using a FluoStar plate reader (BMG Labtech).

**Cell morphology analysis**. PY79 and SH130 cultures were prepared for imaging in CDM and PHMM as described above. Once the cultures had reached OD600 0.4–0.5, Nile Red was added to 200 μL of cells to a working concentration of 1 μg/mL, and incubated at growth temperatures for 5 mins, prepared on agarose microscope slides as described below and cell morphology images recorded. Cell length was then manually determined using ImageJ.

**Western blotting**. Overnight cultures of specified strains were grown overnight in LB at 30 °C. The following morning, cultures were diluted to ~OD600 0.05 and grown at 30 °C until ~OD600 0.6. Cells were harvested by centrifugation and stored at 20 °C for further processing. Samples were normalised based on OD600, re-suspended in 2x SDS protein sample buffer and heated at 80 °C for 10 min followed by centrifugation. ~20 μL lysates were then separated by SDS PAGE in 10% polyacrylamide gel. FtsZ and PBP2B were detected using an FtsZ polyclonal (Sigma, 1:10000 dilution) and a PBP2B polyclonal (Merck, 1:5000 dilution) antibody, respectively, followed by an HRP-conjugated anti-rabbit IgG antibody (Sigma, 1:5000 dilution). Samples were developed using Pierce™ ECL Western Blotting Substrate (Thermofisher) and imaged using an ImageQuant LAS 4000 mini Biomolecular Imager (GE Healthcare).

**Microscopy**. Power density, exposure time and other key parameters are listed for each microscopy experiment in Supplementary Table 4. All imaging was done using either a Nikon N-SIM/N-STORM inverted fluorescence microscope (Fig. 4 and Supplementary Fig. 23) or a custom-built inverted microscope (otherwise).

*Nikon N-STORM*. Cells were illuminated with a 488 nm line from an Argon Ion laser (CVI Melles-Griot). A 100x TIRF objective (Nikon CFI Apochromat TIRF 100XC Oil) was used for imaging and an Andor iXon DU897 EMCCD camera was used, with a 1.5x OptoVar (Nikon) and standard Nikon tube lens, giving an effective image pixel size of 106 nm/pixel. Cells were illuminated via HiLO inclined illumination to minimise background using an objective TIRF module (Nikon N-STORM module) using 488 nm laser excitation.

*Custom microscope.* Cells were illuminated with a 488 nm laser (Obis) and/or a 561 nm laser (Obis) as indicated. A 100x TIRF objective (Nikon CFI Apochromat TIRF 100XC Oil) was used for all experiments except microfluidic VerCINI measurements where a 100x silicone immersion objective (CFI SR HP Plan Apochromat Lambda S 100XC Sil, Nikon) was used for deep focusing. A 200 mm tube lens (Thorlabs TTL200) and Prime BSI sCMOS camera (Teledyne Photometrics) were used for imaging, giving effective image pixel size of 65 nm/pixel. Imaging was done with a custom-built ring-TIRF module operated in ring-HiLO[14] using a pair of galvanometer mirrors (Thorlabs) spinning at 200 Hz to provide uniform, high SNR illumination.

**Nanofabrication**. Micropillars were patterned using electron-beam lithography and etched using reactive-ion etching. Briefly, a silicon wafer was spin coated with AR-N 7700.18 (Allresist GmbH). Features were patterned using a Raith EBPG 5000+ with 56 nm beam size. The unexposed resist was removed with solvent Microposit MF-321 (Dow Chemical Company) and the exposed wafer surface was etched using a Bosch etch process using an AMS 100 I-Speeder (Adixen). The remaining resist was removed using oxygen plasma. Pillars widths ranged from 1.0 to 1.3 μm with height 6.7 μm, spaced 5 μm apart. Final micropillar dimensions were confirmed using SEM.

Micropillar wafers were silanised to allow cured PDMS to be removed without damaging structures. To do this, wafers were coated with (tridecafluoro-1,1,2,2-tetrahydrooctyl) trichlorosilane (abcr GmbH) by vapour deposition.

**VerCINI**. Agarose microholes were formed by pouring molten 6% agarose onto the silicon micropillar array. Patterned agarose was transferred into a Geneframe (Thermo Scientific) mounted on a glass slide, and excess agarose was cut away to ensure sufficient oxygen. Cells at $OD_{600} \sim 0.4$ were concentrated 100x by centrifugation and added onto the agarose pad. Cells were then loaded into the microholes by centrifuging the mounted agarose pad with concentrated cell culture in an Eppendorf 5810 centrifuge with MTP/Flex buckets. Unloaded cells were rinsed off with excess media.

FtsZ treadmilling dynamics were imaged at 1 frame/s (continuous exposure) for 2 mins at 1–8 W/cm². After each field of view was imaged, a multi-slice Z-stack was taken at 500 nm intervals to exclude tilted or improperly trapped cells, in addition to excluding Z-rings that have an adjacent Z-ring 1 μm above or below the imaging plane. VerCINI experiments were performed on cells in a *Δhag* mutant background, preventing undesirable cell rotation within the microholes by disabling flagellar motility.

**Microfluidic VerCINI**
*Device assembly.* Open-topped microhole coverslips were formed from PDMS using the silicon micropillar array (Supplementary Fig. 24). All PDMS reagents and processing equipment were from BlackHoleLab. PDMS elastomer base and curing agent were mixed in a 1:1 ratio and degassed. The degassed mixture was then poured on top of the microhole array and a microscope coverslip was pressed down to form a thin layer. PDMS was cured by baking at 80 °C for 1 h, and the coverslip with patterned PDMS layer was removed from the silicon mould.

Microfluidic chambers were prepared using readily-available components (Supplementary Fig. 24). First, holes were drilled into a glass microscope cover slide using a diamond drill bit (Kingsley North). A flow channel was then made by cutting a strip of double-sided tape using either a scalpel or laser engraver. One side of the double-sided tape was then adhered to the microscope cover slide. Cut pipette tips were then inserted into the drilled holes, and polyethylene tubing (Smiths Medical) was inserted into them. Leakage was prevented by sealing pipette tips and tubing using epoxy.

To load cells into the microholes, PDMS was first rendered hydrophilic by treating with air plasma for 3 min. A 9 mm diameter silicone gasket (Grace Bio-Labs) was then placed over the PDMS microholes. Cells at $OD_{600} \sim 0.4$ were concentrated 100x by centrifugation and pipetted into the silicone gasket. Cells were then loaded into the microholes by centrifuging the microhole-containing coverslip with concentrated cell culture in an Eppendorf 5810 centrifuge with MTP/Flex buckets. The coverslip was covered during centrifugation to prevent evaporation. Unloaded cells were rinsed off with excess media. To ensure adequate loading, this loading-and-rinsing protocol was then repeated.

After cells were loaded into microholes, the silicone gasket was removed and the flow chamber was fully assembled by adhering the PDMS-covered coverslip to the exposed side of double-sided tape on the glass cover slide (Supplementary Fig. 24).

*Imaging.* Due to the relatively thick (~50 μm) layer of PDMS between the coverslip surface and the confined cells, a Nikon silicone immersion objective (CFI SR HP Plan Apochromat Lambda S 100XC Sil, Nikon) was used for imaging due to its high working distance (0.3 mm).

Since the indices of refraction for PDMS and aqueous solution are similar, autofocusing systems relying on the reflection of an IR signal from two interfaces were not adequate due to weak signal. Instead, autofocusing was done using an image-based approach first described by McGorty and co-workers[28] using cross-correlation of brightfield images. Briefly, a separate brightfield imaging pathway was added to the microscope using a 1050 nm IR LED (Thorlabs) and IR-sensitive

camera (UI-1220LE-M-GL, Imaging Development Systems GmbH), and a plugin for Micro-Manager was developed to calculate cross-correlations and move the sample stage (https://github.com/HoldenLab/DeepAutoFocus). At the start of an experiment, a z-stack of IR brightfield images was obtained as a reference. During the experiment, the cross-correlation map between every new IR brightfield image and this reference stack was calculated, and the sample stage was moved to compensate for any drift. To our knowledge this is the first application of IR cross-correlation autofocusing to live cell imaging.

*Fluid control.* Media was flowed through the chamber by a syringe pump (Aladdin-220, World Precision Instruments). To prevent potential leakage, media was pulled through the chamber from fluid reservoirs by operating the syringe pump in withdraw mode. Complete solution exchange occurred within 7 s of initial compound arrival time, estimated using a rhodamine tracer.

**CellASIC experiments**. The microfluidic CellASIC system (EMD Millipore) was used to image constricting cells during treatment with division inhibitors. Cells were loaded into B04A plates and equilibrated in blank media for ~15 min with a pressure of 2 psi (~6 μL/h flow rate) prior to imaging.

In a typical experiment, untreated constricting cells were imaged for 30 min (1 min/frame, 1 s exposure, 1–2 W/cm²) while blank media was flowed using 2 psi. Solution exchange was done by first flowing media with compound using 8 psi (~34 μL/h flow rate) for 15 s, then dropping the pressure to 2 psi for the remainder of the experiment (50–90 min). DMSO treatment alone caused a small but robust increase in constriction time (Supplementary Fig. 25). However, pre-adaptation of cells to blank media containing DMSO eliminated this effect, showing that observed effects of PC19 treatment resulted from the drug and not the solvent that accompanied it (Supplementary Fig. 25).

The arrival time of the compound at the cells was estimated by treating the FtsZ-GFP strain with 10 μM PC19 using our standard flow protocol. PC19 causes the diffuse cytoplasmic FtsZ-GFP to rapidly polymerise into short filaments throughout the cell (Supplementary Videos 10–13), providing a clear signal for compound arrival. From these experiments we determined that compound arrives 3–4 min after fluid exchange.

**Microscopy of horizontal cells**
*Sample preparation.* Coverslips were first cleaned by treating with air plasma for 5 min. Slides were prepared as described previously:[42] Flat 2% agarose pads of either CDM or PHMM were prepared inside Geneframes (Thermo Scientific) and cut down to strips of 5 mm width to ensure sufficient oxygen supply to cells. Cell cultures were grown to $OD_{600}$ between 0.2 and 0.4, when 0.5 μL of cell culture was spotted on the pad. Cells were allowed to adsorb to the pad for 30 s before a plasma-cleaned coverslip was placed on them. After spotting on agarose pads, cells were allowed to equilibrate within the microscope body for 15 min before being imaged. Cells were then imaged either by TIRF microscopy for observation of FtsZ filament dynamics, or ring-HiLO for time-lapse observation, with experimental parameters defined in Supplementary Table 4.

**HaloTag labelling with JF549 dye**. Full labelling of HaloTag-PBP2B was done by incubating strain SH212 with 50 nM JF549 HaloTag ligand for 15 min. Cells were washed once with fresh media before imaging. JF549 HaloTag ligand was a gift from Luke Lavis[35].

**VerCINI data analysis**
*Pre-processing.* Videos were denoised using the ImageJ plugin PureDenoise[15], which is based on wavelet decomposition. Key assumptions of the method are (i) local spatiotemporal similarity within the image (ii) absence of sharp discontinuities, guaranteed by diffraction-limited nature of the image. The denoising algorithm does not make any assumptions on the type of underlying biological structure.

Denoised videos were registered using the ImageJ plugin StackReg[45]. Image stacks of vertically immobilised cells were inspected to exclude tilted cells or out of focus FtsZ-rings, or FtsZ rings where a second Z-ring was closer than 1 μm to the focal plane. Cropped region of interest (ROI) movies containing single in-focus cells were manually selected and exported for analysis. An ImageJ package for VerCINI data processing is available (https://github.com/HoldenLab/Ring_Analysis_IJ).

*Background subtraction, kymograph extraction and quantification.* Custom software for quantitative cytoplasmic background subtraction and kymograph extraction was developed for analysis of VerCINI movies (https://github.com/HoldenLab/ring-fitting2). Fluorescence images of isolated vertical cells were fitted to a joint explicit model for diffuse out-of-focus cytoplasmic background (Gaussian + Cauchy model) plus septal protein signal, modelled as a 12 sectored annulus with each sector of variable amplitude.

$$F(x, y) = \text{Signal}(x, y) + \text{Bg}(x, y) \qquad (1)$$

Signal is calculated in polar coordinates, around the septal ring centre fit parameter $(x_0, y_0)$, and then transformed into Cartesian coordinates. Signal is

defined as

$$\text{Signal}(r, \theta) = A_i(\theta)\exp\left(-\frac{(r - R_0)^2}{2\sigma^2}\right), \qquad (2)$$

where $A_i(\theta)$ defines amplitude of a 12 sectored annulus. Background is defined as

$$\text{Bg}(x, y) = a\exp\left(\frac{-((x - x_0)^2 + (y - y_0)^2)}{2\sigma_{bg1}^2}\right) + b\frac{\sigma_{bg2}^2}{(x - x_0)^2 + (y - y_0)^2 + \sigma_{bg2}^2} + c. \qquad (3)$$

This explicit model allowed accurate sub-pixel location of a circular line profile at the septal leading edge. Explicit subtraction of complex background signal also allowed septal intensity quantification. The background subtracted intensity along the circular line profile was then calculated for each frame and automatically plotted as a kymograph. Kymographs were plotted over the [0, 720°] (i.e. two complete revolutions around the line profile) for ease of visualisation, specifically to allow straightforward visualisation of treadmilling filaments crossing the 360°, 0° boundary. For each cell, the Z-ring diameter was determined from the fitted model in the kymograph extraction step. Median septal intensity was calculated as the median value of all kymograph pixels from the first 60 image frames.

*FtsZ filament treadmilling analysis.* We found that automated methods for in vitro FtsZ filament speed measurements, either after Dos Santos Caldas and coworkers[46] or using a similar in-house algorithm, are currently insufficient for FtsZ filament speed analysis in either horizontal or vertical cells compared to manual analysis, because at cellular filament density they currently perform poorly at detecting immobile filaments, break the complete trajectories of treadmilling filaments into multiple shorter trajectories, and cannot reliably process images from dense Z-rings in horizontal cells.

We therefore performed manual measurement of filament speed and processivity by kymograph analysis, annotating filaments as lines in ImageJ and then measuring the angle via ImageJ script. As the intensity range across kymographs is large, manual detection and analysis of filaments was aided by applying a ridge filter[47] to the kymograph which detects ridges in the image, independent of absolute intensity. The ridge filter consists of a Gaussian blur operation to set the feature scale and suppress noise (here 2 pixels, 130 nm), followed by calculation of the major eigenvalue of the Hessian matrix for each pixel, implemented as an ImageJ macro in the package listed above. Joint annotation of the ridge filtered and raw intensity kymograph was performed, allowing sensitive annotation of both bright and dim filaments (Supplementary Fig. 26).

**Horizontal cell image analysis**
*Pre-processing.* Images were denoised and registered as described above.

*Cell segmentation and tracking of division protein localisation.* MicrobeJ[48] was used to track individual FtsZ-/PBP2B-rings and export their centroid coordinates for analysis. To do this, image backgrounds were first subtracted using a sliding paraboloid window of 50 pixels and all frames were converted to 8-bit for easier thresholding.

Two approaches were used to identify division protein localisation, depending on the experiment. For measurements of septal constriction rate only (Fig. 4a), which requires only detection of bright dense FtsZ/PBP2B rings, a simple approach of intensity-based threshold detection was applied. In MicrobeJ a threshold was selected manually for each field of view using MinError. Additional area and length thresholds (Area: [0.08 Inf] μm², Length: [0.02 1.8] μm) were established to exclude objects in fields of view that were not rings. Tracks shorter than seven frames were excluded. Data were then exported for analysis in MATLAB.

For analysis of FtsZ ring dynamics over the entire division process, from nascent Z-ring assembly to late constriction (Fig. 1), it was necessary to additionally robustly detect diffuse, patchy FtsZ-rings. We trained the machine learning segmentation tool Ilastik[49] to detect cell outlines with fluorescence images of cells labelled with FtsZ. For detection of diffuse Z-rings, we applied a large Gaussian blur filter to the images, and then multiplied this by the Ilastik-segmented binary image, which had the effect of separating any overlap of Z-rings between adjacent cells. Intensity threshold-based detection of Z-rings was then performed in MicrobeJ as above.

*Quantification of division protein septal localisation.* Custom MATLAB software was developed to measure the diameters and intensities of Z- and 2B-rings obtained from either MicrobeJ or Ilastik. Each frame of each individual ring was first segmented using a multi-level threshold to distinguish the septum, cytoplasm and background. The septal axis and the cell axis were identified using the segmented image, and the image was rotated such that pixels were aligned along these axes. Line profiles of Z- and 2B-rings were then obtained by averaging pixel values along the septal axis ±2 pixels to reduce noise. Diameters were then obtained by fitting line profiles to a toy model that describes a circle of uniform intensity rotated to be orthogonal to the axis of the imaging plane ('Tilted Circle Model', Supplementary Note 1 and Supplementary Fig. 27). Septal density was calculated as the total intensity along the lateral line profile, divided by the ring circumference, as calculated from the constriction rate analysis (below) to reduce single frame noise.

Septal density was used in place of total intensity in order to compare to the septal intensity measurements obtained during the VerCINI experiments. Axial thickness was estimated by fitting an axial line profile to an analytical super-Gaussian and calculating the full-width half maximum.

*Evaluation of percentage Z-rings condensed and constricted post-treatment.* Since PC19 treatment causes bright fluorescent foci to appear randomly throughout the cell in strains where FtsZ is labelled, tracking software fails to consistently follow most Z-rings post-treatment in the SH130 and SH212 strains. Furthermore, measurements of Z-ring thickness and diameter were inaccurate due to the proximity of fluorescent foci. Since it was typically straightforward to observe whether Z-rings condensed and/or constricted post-PC19 treatment, this was evaluated manually.

*FtsZ filament treadmilling analysis from TIRF microscopy data.* Images were denoised and registered as above. TrackMate[50] was used to help identifying treadmilling FtsZ filaments outside of dense Z-rings. A line ROI was drawn perpendicular to the cell axis where treadmilling filaments were identified. Kymographs were calculated along this line profile FtsZ filaments trajectories in the kymograph were marked with a line ROI (Supplementary Fig. 23). Treadmilling speed for each filament trajectory was calculated from the angle of the line ROIs.

**Constriction rate analysis.** Previous models for septal constriction have either been ad hoc[51] or tailored to Gram-negative cell division based on remodelling of a hemispherical cap[52]. We derived a simple model for Gram-positive constriction, assuming (i) outside-in synthesis of a flat plate at mid-cell and (ii) total PG synthesis rate around the septal leading edge is constant (Supplementary Note 2)

$$d(t) = \left\{ d_0, t < t_0 \sqrt{d_0^2 - \frac{4k}{\pi}(t - t_0)}, t \geq t_0. \right. \qquad (4)$$

Where $d_0$ is the initial Z- or 2B-ring diameter, $t_0$ is the time constriction begins, and $k$ is the rate at which area is added to the septal plate. This model fitted measured constriction trajectories well, and the estimated time of constriction completion corresponded closely to completion times independently estimated from PBP2B departure in wild type cells (Supplementary Fig. 28).

We measured constriction rates in the mNeonGreen-PBP2B strain, since FtsZ-GFP mislocalises post-treatment with PC19 and makes it difficult to track large numbers of septa. A small amount of mNeonGreen-PBP2B signal remained at mid-cell after constriction had finished. This excess signal was excluded from the rate analysis by cutting the 2B-ring data after this drop in intensity. We obtained constriction time distributions for PC19-treated cells by fitting only to post-treatment data.

For comparison to previous observations we converted the observed constriction rates to an effective constriction time $t_{\text{eff}}$, the total amount of time it would take for an unconstricted septum to constrict completely (Methods) given the fitted rate $k$: $t_{\text{eff}} = 2\pi \cdot \frac{d_0^2}{k}$. The average diameter of the unconstricted septum $d_0$ was estimated as $1100 \pm 100$ nm (mean, SD) for the mNeonGreen-PBP2B based on the background fluorescence signal in the cell.

**Statistics.** Data were analysed using estimation statistics, which focusses on magnitude and robustness of effect size using DABEST[53] (Data Analysis with Bootstrap Coupled Estimation), rather than just statistical significance/$p$ value. Averages reported were median values unless otherwise indicated.

The basic principle of a DABEST plot is to calculate and plot the average (here median) difference between two conditions, together with the 95% confidence interval on that difference. This has the advantage over null hypothesis significance testing (NHST) of visualising not just the probable reproducibility of a difference between two conditions, but also the estimated magnitude of the difference between the two conditions. This is useful because it allows assessment of whether magnitude of difference is physically/ biologically meaningful, rather than just reproducible. For comparison to NHST analysis, we note that a median difference DABEST plot where the 95% CIs do not overlap the 0—difference axis would indicate rejection of the null hypothesis that the two population medians are equal at $p = 0.05$ significance level.

All effect size estimates were calculated based on difference of medians. 95% confidence interval of the median, or of the difference of medians, was estimated by bootstrapping (https://github.com/HoldenLab/violinplusDABEST-Matlab) and indicated with square brackets []. Interquartile range was indicated by IQR. Thick error bar lines in Violin plots indicate IQR, thin lines indicate adjacent vales. Error bars in DABEST plots indicate 95% CI of median difference, estimated by bootstrapping. Sample size, indicating number of filaments/Z-ring, cells, microscope fields of view, technical and biological replicates, as appropriate, is presented for each dataset in Supplementary Table 3.

**Simulations of FtsZ filament dynamics.** Simulated time-lapse movies of FtsZ filament dynamics in vertically immobilised cells were generated. Source code for simulations is available (https://github.com/HoldenLab/ring-simulator). Filament speed and lifetime were sampled from distributions obtained from experiments via bootstrap resampling. FtsZ filament distributions were taken from cells expressing

FtsZ-GFP in the nascent Z-ring cell division state. Sub-diffraction-limited treadmilling filaments were simulated on a diffuse Gaussian–Cauchy cytoplasmic background described above. Filament intensity, background amplitude and shape parameters all matched average parameters experimentally measured in the Ver-CINI images of nascent Z-rings. Photon noise and manufacturer specified camera gain and read noise were added to the images.

The number of filaments in each simulated Z-ring was drawn from a Poisson distribution with average filament density equal to observed nascent, mature, 2x mature or 12x mature Z-ring density of (1.3, 3.3, 6.5, 37.5 filaments/ Z-ring/ frame). Filament density of real nascent Z-rings was calculated directly from observed number of manually traced filaments. Filament density in mature Z-rings was estimated by multiplying the estimated nascent Z-ring density by 2.6, the ratio of total Z-ring intensity for mature and nascent Z-rings (Supplementary Fig. 8b).

**Step finding analysis of Z-ring axial thickness data**. Transitions between a decondensed and condensed Z-ring for SH130 cells expressing FtsZ-GFP Z-rings was measured using a step finding algorithm to detect either one or two states in time series Z-ring axial thickness, using a global change point detection algorithm[54] implemented in the MATLAB function findchangepts(). State detection was performed based on changes in standard deviation of the time series, with an additional threshold that detected axial thickness states should differ by at least 50 nm.

**Analysis of Z-ring axial thickness data for FtsZ(D213A) mutant**. Although the step-finding method described above accurately located the decondensed-to-condensed transition in wild-type cells, we found that this method was inadequate for the FtsZ(D213A) mutant since these Z-rings were in a partially-condensed state and did not show a similar transition as wild-type. Instead, we considered these Z-rings as having low thickness pre-constriction if their thicknesses dropped below a threshold value for at least one frame (1 min) prior to the start of constriction. The threshold was chosen as 1σ above the mean thickness for condensed wild-type cells (356 nm; Supplementary Fig. 17).

**Statistics and reproducibility**. All sample sizes and number of experimental replicates can be found in Supplementary Table 3.

**Reporting Summary**. Further information on research design is available in the Nature Research Reporting Summary linked to this article.

## Data availability

Data supporting the findings of this manuscript are available from the corresponding author upon reasonable request. A reporting summary for this Article is available as a Supplementary Information file. Source data are provided with this paper. Furthermore, source data for all figures presented in the paper and Supplementary Information, as well as raw movie files for all Supplementary Videos are available at https://data.ncl.ac.uk/projects/FtsZ_treadmilling_is_essential_for_Z-ring_condensation_and_septal_constriction_initiation_in_Bacillus_subtilis_cell_division/92465.

## Code availability

Custom software is available on the Holden lab GitHub page or Zenodo:
https://github.com/HoldenLab/DeepAutoFocus[55]
https://github.com/HoldenLab/Ring_Analysis_IJ[56]
https://github.com/HoldenLab/ring-fitting2[57]
https://github.com/HoldenLab/violinplusDABEST-Matlab[58]
https://github.com/HoldenLab/septal_constriction_analysis[59]
https://github.com/HoldenLab/ring-simulator[60]

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

## Acknowledgements

We thank Henrik Strahl (Newcastle) for helpful discussions, Ling Wu and Jeff Errington (Newcastle) for strains and antibodies, Luke Lavis (HHMI) for Janelia Fluor dyes and Ethan Garner (Harvard) for strains. S.H., K.D.W., C.J. and E.K. acknowledge funding support by a Newcastle University Research Fellowship and a Wellcome Trust & Royal Society Sir Henry Dale Fellowship grant number [206670/Z/17/Z]. N.T. supported by a UK Engineering and Physical Sciences Research Council doctoral studentship. R.H. and P.A. funded by grants from the UK Biotechnology and Biological Sciences Research Council [BB/M022374/1], the UK Medical Research Council [MR/K015826/1] and Wellcome Trust [203276/Z/16/Z]. C.D. and K.W. acknowledge funding support by ERC grant SynDiv [669598] and the Netherlands Organisation for Scientific Research (NWO/OCW) as part of the Frontiers of Nanoscience and BaSyC programs.

## Author contributions

K.D.W., C.J., N.T. and E.K. acquired data. K.D.W., C.J., N.T., E.K. and S.H. analysed data. K.D.W., C.J., S.H. and C.D. developed the VerCINI method. K.D.W. performed nanofabrication, supervised by C.D. S.H., K.D.W., P.A. and R.H. wrote software. C.J., E.K., N.T. and K.D.W. created and characterised bacterial strains. Y.C., S.H. and K.D.W. built the custom microscope. S.H. and C.D. directed the research. K.D.W., S.H. and C.J. wrote the manuscript with input from all authors.

## Competing interests

The authors declare no competing interests.
