## [Peer Review File · Nature Communications]

Reviewer #1 (Remarks to the Author):

The manuscript "FtsZ treadmilling is essential for Z-ring condensation and septal constriction initiation in bacterial cell division" by Whitley and colleagues aims at understanding the role of FtsZ treadmilling in the process of septal cell wall synthesis and cell division. The authors use a modified version of the previously described imaging technique using micro-fabricated "holes" in which bacterial cells are placed and imaged in vertical direction, thereby allowing to visualize Z-rings. The main findings described here are that FtsZ treadmilling is important for Z rings to coalesce and form a dense Z ring at the site of septation and subsequent recruitment of other division components. After initiation of septal cell wall synthesis the treadmilling becomes dispensable, but somehow functions in accelerating septal constriction.

The topic is of high importance and the molecular mechanisms behind bacterial cytokinesis have gained a tremendous attention in recent years. Work from the author's lab including the newly presented data increase our knowledge about this important aspect of bacterial self-organization. Thus, the topic and the approach are well suited for publication, but I suggest several points that should be considered in a revision. A main point of criticism is the unclear use of different FtsZ-FP fusions.

Points to consider:

1. The authors use different strains in their study. The strain used in most of the experiments is SH130. This strain is constructed using a strain constructed by the Levin lab in 1999 (the original paper cites this strain as PL642 not PAL642). This strain contains an allelic replacement of *ftsZ* with an *ftsZ-gfp* allele. The authors described this strain as temperature sensitive. This is in line with the observation here that cell length phenotypes are mild at 30°C but relatively strong at 37°C (Suppl. Fig. 2). Some of the experiments are however done at 37°C and under these conditions, I am not sure if FtsZ-GFP is a suited fusion protein that can reliably report FtsZ dynamics. There are several reports about fully functional FtsZ-FP fusions for the *E. coli* FtsZ and an analogous strategy for the *Bacillus* FtsZ might prove useful.
2. The authors state differences in growth rate at several places in their work. Unfortunately, there is not a single value given for growth rates (nor growth curves from which these values could be deduced).
3. Figure 1 h shows FtsZ speed in nm/s. Basically this plot shows that in nascent septa FtsZ is often immobile (speed close to zero) and in mature and late constricting septa the speed is more or less similar (and importantly similar to the dynamic ones at early phase (between 20-30 nm/s). Is there an explanation for the high number of immobile FtsZ early in septation?
4. In Figure 3 C the authors show co-localization of FtsZ-GFP and a Halo-tagged PBP2B. The images show two distinct septa in which the upper one is said to have less PBP2B and not constrictive (at least I interpret the notion "did not divide" (typo btw in "divide") like this). The FtsZ ring seems nevertheless to constrict. How would that be possible? Furthermore, strain 2020 contains an amyE integration of a P-xyl inducible GFP-FtsZ fusion. Thus, this *ftsZ* allele encodes for a C-terminal fusion of FtsZ and is therefore not necessarily comparable to the N-terminal fusions used in other experiments. Was there a specific reason for using different *ftsZ*-fp alleles?
5. I miss a statistical analysis of all plots showing dynamics, cell length etc.
6. At several cases (paragraph starting at line 96) the authors say that FtsZ treadmilling is cell cycle regulated. I do not quite understand this argumentation. Regulation implies to me that there are differences in the speed of treadmilling or the percentage of FtsZ treadmilling filaments. Further, cell cycle in my view is the complete cycle from replication initiation to termination and subsequent

division. This cell cycle has not been looked at here, or do I miss something?

7. The authors use the GTPase inhibitor PC190723 to stop FtsZ treadmilling. Control experiments show that the effect on treadmilling is immediate, however, in the original publication (Haydon et al., 2008) addition of PC190723 leads to a dispersed, punctate localization of FtsZ. Benzamides were shown to enhance FtsZ bundling and therefore I am not sure if PC addition is ideally suited to control treadmilling behavior. This is in particular important since in *E. coli* effects are described where PC might have targets other than FtsZ polymerization. I therefore like the use of the FtsZ treadmilling mutants such as D213A (I assume this is the equivalent mutant to D212G described in *E. coli* (Yang et al., 2017)). Use of this mutant should be explained better in the text to make this point clear.

8. Suppl. Figure 1: Indicate how much lysate (total protein) was loaded. Ideally provide internal control and show blots for all strains used in this study that harbor FP fusions.

Reviewer #2 (Remarks to the Author):

Overall this is a very nice manuscript that uses an elegant microfluidic imaging platform to visualize the motion of FtsZ filaments in *Bacillus subtilis*. The platform enables quantitative analysis of FtsZ motion, septal constriction, and response of this process to chemical perturbation to provide some basic insights into the role of FtsZ treadmilling in cell division.

The videos in the supplement are very impressive, and the image processing, analysis and extraction of quantitative information was adequately described, reasonably justified, and transparent. The methods were cited appropriately when used in previous work, allowing reasonable follow up by the reader.

Some questions/concerns that I would like addressed...

1. In the initial description of the 3 "distinct" phases of z-ring stages (nascent, mature, constricting), I was not fully convinced based on figure 1B that these phases were quite distinct. Visually there is a marked change in the slope of the curves in 1B when cells begin constricting, but the use of a cutoff of 400nm to distinguish between nascent and mature seems weak. 1C is colored to show 3 clusters of cells, but it is not clear how distinct these clusters really are.

In all, the use of these classifications as descriptive and speculative seems fine for the sake of discussion. However, a description of these as "distinct" phases begs for stronger justification. Also please consider the next comment below and how it might impact distinction between stages etc.

2. Throughout the manuscript overall thickness and diameter measurements are used to develop scatter plots and cluster cells. This would benefit from a clearer picture of variation in cell size and diameter for the normal population of cells. For example the cells are described as "normal" versus "mildly elongated" at different temperatures (82-83). What was the actual difference in length and how did this vary for the population? Can diameters, be expressed as % change in diameter for each cell? Were you able to capture the actual diameter or starting diameter of a cell versus the z-ring diameter. Expressing normalized % diameter would likely reduce some of noise in the scatter plots and collapse some of the groupings. (i.e. a cell that appears to have a smaller diameter ring may just be a smaller cell and not really be further along in the ring formation process.)

Alternatively, providing a description of the size distribution for the population of cells used and showing that the population does not vary in diameter or length significantly compared to the variation in diameter seen in the z-rings would provide adequate justification to not perform a normalization. (i.e. if everything is divided by the same number...no need to normalize)

3. Please discuss the use of agarose versus PDMS in different stages more clearly. Describe why the change was made and how the use of the hydrogel versus the PDMS might impact transport of nutrients, cell health and response.

4. You conclude that the motion of FtsZ leads to more collisions and condensations. This is fairly obvious or intuitive, but it is nice to visualize. However, there seems to be a lack of discussion as to why this occurs in the center of the cells. Is there something in the curvature of the cell ends that drives the FtsZ to the center of the cell, causing a split? Is there molecular crowding at the ends of the cells that drive FtsZ to the center? Is this something that could be modeled and addressed more deeply in discussion? By itself, the finding that motion leads to more collisions, leads to condensation, seems like an obvious statement and rather uninteresting. What drives this to occur at the center of the cell and allow an even split?

Overall this is a very nice manuscript and highlights a nice method for visualizing molecular processes in bacterial cells. I recommend publication of the manuscript once the above concerns/comments are addressed.

Regards,
Scott T. Retterer

Reviewer #3 (Remarks to the Author):

In the manuscript "FtsZ treadmilling is essential for Z-ring condensation and septal constriction initiation in bacterial cell division", Whitley et al. use novel and existing microscopy techniques to resolve differences in FtsZ dynamics throughout the stages of the cell cycle in *Bacillus subtilis*. The authors determined that *B. subtilis* constriction can occur independently of FtsZ polymer dynamics after constriction activation. This observation was made possible by creative use of "micropillars" to vertically position and image *B. subtilis* cells both in agarose "microwells" and in a microfluidics device under flow. Positioning cells vertically, de-noising the images by mathematically enhancing the signal-to-noise ratio, and using more accurate models for calculating constriction rate have all contributed to improved resolution of how FtsZ dynamics and treadmilling contribute to constriction. These methods, combined with the use of the established FtsZ inhibitor, PC190723 (PC19), demonstrate that FtsZ treadmilling is halted in cells within a few seconds of treatment with PC19. Their approach allows the authors to classify constriction timing and determine that when FtsZ dynamics are halted with PC19, cells will only constrict if constriction has already initiated. Lastly, the authors observe that FtsZ dynamics are independent of the temperature or richness of the medium, suggesting that the rate-limiting step of cell division in *B. subtilis* is growth rate rather than FtsZ dynamics.

The major strength of this study is the novel microscopy techniques employed, which arguably provide the best resolution of FtsZ dynamics and constriction yet presented. Despite the beautiful

use of these techniques, the novel mechanistic insights that this manuscript presents are bit more limited. The finding that FtsZ dynamics are not required for constriction in *B. subtilis* after initiation of constriction echoes a similar finding in *Staphylococcus aureus* (Monteiro et al 2018 Nature). The major difference between the two may be that dynamic FtsZ still contributes to an optimal constriction rate in *B. subtilis*, but the contribution of FtsZ dynamics to constriction rate in *B. subtilis* was also previously established (Bisson-Filho et al 2017 Science). The conclusion that FtsZ treadmilling contributes to “Z-ring condensation” is not as well-developed here, and it has long been established that FtsZ dynamics are important for Z-ring formation and structure at midcell. Overall, the paper is well-written, the microscopy is really impressive and allows the most comprehensive view of *B. subtilis* Z-ring dynamics and function yet presented, but the novelty of the mechanistic insights may be more limited.

Specific comments:

1. Throughout, the authors use PC19 to impact/prevent FtsZ dynamics. However, it's important to note that PC19 does not specifically stop treadmilling, but impacts FtsZ GTPase activity and filament structure (Haydon et al 2008 Science, Adams et al 2011 Mol Micro). Use of PC19 therefore does not allow one to specifically comment on the role of treadmilling, but of FtsZ dynamics/structure overall. This should be mentioned and elaborated upon in the context of the findings of this paper that involve PC19 use.
2. Title: The paper does not address “bacterial cell division”, it addresses *B. subtilis* cell division. This is important to clarify since there are well-documented differences in Z-ring dynamics and their relationship to cell wall metabolism between *B. subtilis* and other bacterial species (e.g. *E. coli* and *S. pneumoniae*).
3. Throughout, the authors allude to the “movement” or “motility” of treadmilling FtsZ filaments. Since treadmilling is sometimes a difficult concept for non-experts to understand, it's important to be very clear that it is apparent motion/motility/movement and that single molecules of FtsZ do not move.
4. Line 118 and Figure 1eii: It is hard to determine when two clusters collide and become temporarily arrested. Can you highlight this in the figure?
5. It is surprising that nascent rings appear to have reduced treadmilling compared to mature and late rings. There are numerous studies in a range of bacteria showing that Z-ring localization is highly dynamic early in the cell cycle and becomes more stable later. This would imply that FtsZ filaments are at least as dynamic in nascent rings as in mature rings, and perhaps more so. Mechanistically, it's also difficult to imagine how FtsZ filaments would be more stable in nascent rings. Could this be an artifact of the difference in density of clusters in nascent vs mature rings? That is, as rings become denser, it becomes much more difficult to pick out individual FtsZ clusters. Perhaps non-moving clusters are obscured and under-counted in mature and constricting rings? If the authors don't think this is a technical artifact, some further explanation is required for how/why nascent rings might not treadmill. Another possibility is that, in fact, the nascent FtsZ structures are still very dynamic, but do not undergo treadmilling (i.e. if you performed FRAP on nascent rings, they would show similar dynamics of recovery as mature rings). Related, Line 122: “drop in the number of immobile FtsZ filaments” could just be a drop in the ability to detect immobile FtsZ filaments.
6. One of the major conclusions of the paper is that FtsZ treadmilling is required for Z-ring condensation, but the data presented to support this are fairly limited. In figure 3b and supplementary figure 8, how was “condensed” vs “not condensed” determined?
7. Line 117: why is it surprising to observe clusters of FtsZ colliding and fusing/aggregating? Line 120: it may not be “treadmilling” of FtsZ that specifically drives filament encounters, but FtsZ dynamics in general.

8. It would be lend confidence and specificity to the conclusions if the authors were able to complement the PC19 treatment with other ways of perturbing the Z-ring and its dynamics. For example, how is PBP2B intensity impacted in strains producing FtsZ GTPase mutants that have slower treadmilling? What happens to constriction if Z-rings are disassembled (e.g. with expression of MciZ)?
9. Line 221: since the role of FtsZ dynamics in formation of a focused midcell ring is well-established, it's not clear how "novel" the role of Z-ring dynamics in "condensing" the Z-ring is. PC19 was previously shown to disrupt and extend Z-ring structures, e.g. in Adams et al 2011 Mol Micro.
10. Line 94: It would be helpful to state the classification criteria (listed on lines 328-331 in the supplement) for the different ring states in the main texts for ease of reference and clarity.
11. Line 111: More accurate to say figure 1e-h
12. Line 122-126: Data is in Figure 1h so that should be referenced.
13. Lines 131-132: The data in figure 1 do not show that "FtsZ treadmilling [promotes] Z-ring condensation prior to septal constriction initiation".
14. Line 218-220: It's not clear what the authors mean by "FtsZ treadmilling [guides] septal synthases". Directs their dynamic localization? Regulates their activity?
15. Line 283: This study does not investigate FtsZ lateral interactions at all, so conclusions about lateral interactions should be avoided.
16. Supplemental Line 21: "... at indicated temperature." Add 'the'
17. Supplemental figure 6 appears to suggest there is a larger fraction of nascent rings measured based on the VerCINI method than what is measured based on the horizontal cell method. Do they have any comment on this?
18. Supplementary figure 8: Is it significant that HT-PBP2 expressing cells are more likely to have condensed rings during PC19 treatment? Could this be the tag contributing to aggregation/an increase in density during division? Any explanation?
19. Supplementary note 1: It would be helpful to provide a diagram if possible to help with the explanation.

Author response:

We would like to thank all the reviewers for their helpful comments and for their careful reading of the manuscript. We have revised the manuscript, collected new data, and incorporated new analysis into the text to address all the concerns raised by the reviewers.

Below we respond to the criticisms raised by the reviewers point by point.

We have noted in **bold** changes made to the manuscript.

Reviewer 1 (Remarks to the Author):

The manuscript "FtsZ treadmilling is essential for Z-ring condensation and septal constriction initiation in bacterial cell division" by Whitley and colleagues aims at understanding the role of FtsZ treadmilling in the process of septal cell wall synthesis and cell division. The authors use a modified version of the previously described imaging technique using micro-fabricated "holes" in which bacterial cells are placed and imaged in vertical direction, thereby allowing to visualize Z-rings. The main findings described here are that FtsZ treadmilling is important for Z rings to coalesce and form a dense Z ring at the site of septation and subsequent recruitment of other division components. After initiation of septal cell wall synthesis the treadmilling becomes dispensable, but somehow functions in accelerating septal constriction.

The topic is of high importance and the molecular mechanisms behind bacterial cytokinesis have gained a tremendous attention in recent years. Work from the author's lab including the newly presented data increase our knowledge about this important aspect of bacterial self-organization. Thus, the topic and the approach are in well suited for publication, but I suggest several points that should be considered in a revision. A main point of criticism the unclear use of different FtsZ-FP fusions.

We note the positive comments and thank you for constructive criticism. We have carefully addressed all points raised and in particular have clarified the use of different FtsZ-FP fusions in our responses to comments 1 and 4.

Points to consider:

1. The authors use different strains in their study. The strain used in most of the experiments is SH130. This strain is constructed using a strain constructed by the Levin lab in 1999 (the original paper cites this strain as PL642 not PAL642). This strain contains an allelic replacement of ftsZ with an ftsZ-gfp allele. The authors described this strain as temperature sensitive. This is in line with the observation here that cell length phenotypes are mild at 30°C but relatively strong at 37°C (Suppl. Fig. 2). Some of the experiments are however done at 37°C and under these conditions, I am not sure if FtsZ-GFP is a suited fusion protein that can reliably report FtsZ dynamics. There are several reports about fully functional FtsZ-FP fusions for the E. coli FtsZ and an analogous strategy for the Bacillus FtsZ might prove useful.

The reviewer raises a valid point about the temperature sensitivity of the SH130 strain we used. Since the cells display an elongation phenotype at 37°C, it is reasonable to suspect that the measured dynamics of FtsZ-GFP may not fully represent those of wild-type FtsZ under these conditions. We previously ensured that all measurements of FtsZ-GFP treadmilling using SH130 (Figs. 1 and 2) were performed at 30°C, where there is little phenotypic difference with wild-type cells (Supp. Fig. 2). The only experiment we performed with SH130 at 37°C was the effect of PC19 on condensation and constriction (Fig. 3a-b). To ensure these results were not an artefact of the

temperature-sensitivity of the strain we also previously performed this experiment at 30°C (Supp. Fig. 9; revised: 14), but we acknowledge that the data in this plot was rather scarce. **To support this conclusion, we have collected more data with this strain at 30°C and added it to the figure (revised Supp. Fig. 14).** With these new data, no results in the paper depend on the use of this strain at 37°C.

Additionally, our conclusion that FtsZ treadmilling is required for Z-ring condensation and constriction initiation is supported by independent measurements in a distinct merodiploid strain harbouring an N-terminal GFP-FtsZ fusion (strain SH212; Fig. 3c-d, Supp. Figs. 8, 10 (revised: 13, 18). This (2-colour) strain was constructed via backcross from a previously-characterized GFP-FtsZ fusion strain (Adams et al. 2011), and the fusion protein is expressed at low levels in our experiments so as to be minimally perturbative to cell physiology. **We have now clarified this point and elaborated on the rationale and construction of the SH212 strain (GFP-FtsZ, HaloTag-PBP2B) in the Results section, as previously it was unclear that we had already performed measurements on two distinct FtsZ-FP fusions (L258-263):**

“Since fully-labelled FtsZ slightly perturbed cell physiology (Supplementary Figure 2) and full labelling of FtsZ was unnecessary for this assay, for the labelled FtsZ we decided to use a well-characterized merodiploid harbouring an N-terminal GFP-FtsZ fusion²² that could be expressed at a low non-perturbative level. We therefore constructed a 2-colour strain (SH212, Methods) expressing GFP-FtsZ from an ectopic locus and HaloTag-PBP2B from its native locus.”

We note the reviewer’s comment about the sole copy FtsZ-FP construct in *E. coli* (Moore et al. *J Bac* 2017 & derivatives). We have previously tried the Moore FtsZ side-loop cloning strategy but did not yet produce a more functional fusion than the PL642 strain. We wish to point out that all the fusions presented by Moore and coworkers have a similar moderate elongation phenotype as the established *B. subtilis* PL642 FtsZ-GFP fusion, and the phenotype of the *E. coli* strains at 45°C temperature was not reported. While the Moore *E. coli* FtsZ fusion can support growth as the sole cellular source and was a major step forward for FtsZ fusion functionality in *E. coli*, its functionality appears broadly similar to the *B. subtilis* PL642 fusion.

We have renamed the strain PAL642 to PL642 for consistency with the original 1999 paper in which it was first described, thank you for pointing this out.

2. The authors state differences in growth rate at several places in their work. Unfortunately, there is not a single value given for growth rates (nor growth curves from which these values could be deduced).

We have now provided growth curves for the ME7 strain under all growth conditions described in the text as Supp. Fig 3c along with a plot of the effect of growth rate on constriction rate as Supp. Fig. 3d. We have also now provided a table of doubling times as Supp. Table 1. Furthermore, we have also provided growth curves for relevant strains under two conditions in Supp. Fig. 3a-b.

3. Figure 1 h shows FtsZ speed in nm/s. Basically this plot shows that in nascent septa FtsZ is often immobile (speed close to zero) and in mature and late constricting septa the speed is more or less similar (and importantly similar to the dynamic ones at early phase (between 20-30 nm/s). Is there an explanation for the high number of immobile FtsZ early in septation?

There are two likely mechanisms for immobile FtsZ filaments. 1) When the local FtsZ monomer concentration is low, filaments may depolymerise from the minus end without fast growth on the

plus end, i.e. only binding and turning over at the membrane rather than treadmilling. 2) Accessory proteins or membrane anchors may modulate the turnover of FtsZ from the +/- filament ends, as observed *in vitro* (Loose et al. *Nat Cell Biol* 2014) where FtsZ-ZipA produced largely static filaments but FtsZ-FtsA supported FtsZ treadmilling. **We have now added speculation to this effect in the Discussion section (L342-344):**

“While the mechanism of FtsZ filament motility regulation requires further study, *in vitro* measurements of FtsZ dynamics suggest that immobile FtsZ filaments could result from low septal FtsZ concentration or regulation by accessory/anchor proteins⁹.”

The mechanism of FtsZ filament motility regulation is an important but quite substantial question which we aim to follow up in future work.

4. In Figure 3 C the authors show co-localization of FtsZ-GFP and a Halo-tagged PBP2B. The images show two distinct septa in which the upper one is said to have less PBP2B and not constrictive (at least I interpret the notion “did not divide” (typo btw in “divide”) like this). The FtsZ ring seems nevertheless to constrict. How would that be possible?

We understand the reviewer’s confusion regarding the “Low 2B” case in Fig. 3c. The Z-ring in that case decreased in fluorescence *intensity* over time around 10-20 min, but did not decrease in *diameter* (in contrast to the “High 2B” signal, where diameter clearly decreased). It was therefore considered a non-constricting cell. Due to low labelling density resulting from the strain expressing GFP-FtsZ and HaloTag-PBP2B from inducible promoters (SH212; Fig. 3c), these experiments have significantly lower signal-to-noise and suffer more from photobleaching than the strain expressing FtsZ-GFP from the native promoter (SH130; Fig. 3a).

Upon looking back at Fig. 3c we recognise that the example provided is difficult to interpret. This example was selected because the initial diameter and GFP-FtsZ intensity of the “Low 2B” case were very similar to those in the “High 2B” case. However, we did not return to this point of comparison in the text, and so it is unnecessary. **We have therefore replaced the “Low 2B” example with a ring displaying a higher initial GFP-FtsZ signal that does not fade away significantly post-PC19 treatment, and is hence visually more comparable to the non-constricting Mature cell in Fig. 3a.**

Additionally, to better clarify the results, **we have now replaced the terms “Did not divide” and “Divided” with “Did not constrict” and “Constricted”, respectively, throughout the manuscript.**

Furthermore, strain 2020 contains an amyE integration of a P-xyl inducible GFP-FtsZ fusion. Thus, this ftsZ allele encodes for a C-terminal fusion of FtsZ and is therefore not necessarily comparable to the N-terminal fusions used in other experiments. Was there a specific reason for using different ftsZ-fp alleles?

The N-terminal fusion was used as a dilute ectopic label in 2-colour imaging experiments where the primary label for quantification was the PBP2B-fusion. In Fig. 3c-d GFP-FtsZ was used to measure Z-ring dimensions prior to PC19 treatment and to assess whether Z-rings condensed or constricted post-treatment. It was not used to quantify FtsZ treadmilling speeds or septal constriction rates. Low level merodiploid expression of GFP-FtsZ minimizes perturbation to cell physiology and is thus useful where quantification of total FtsZ filament dynamics is not required.

However, we have noticed that we misannotated this as FtsZ-GFP (i.e. C-terminal fusion) rather than GFP-FtsZ. **We have now corrected this throughout the manuscript and explicitly stated that this strain harbours an N-terminal fusion.**

5. I miss a statistical analysis of all plots showing dynamics, cell length etc.

Statistical analyses of plots are already present in estimation-statistics-based DABEST plots, which were used in preference to significance testing (Fig. 4, Supp. Fig. 7, 13, 16 (revised: 9, 20, 24), details of analysis in Methods:Statistics).

We have now provided DABEST plots comparing FtsZ treadmilling speeds between chemical perturbations (Fig. 2c) as Supp. Fig. 12. We have also now provided a DABEST plot comparing cell lengths of the SH130 strain under different growth conditions to Supp. Fig. 2.

6. At several cases (paragraph starting at line 96) the authors say that FtsZ treadmilling is cell cycle regulated. I do not quite understand this argumentation. Regulation implies to me that there are differences in the speed of treadmilling or the percentage of FtsZ treadmilling filaments. Further, cell cycle in my view is the complete cycle from replication initiation to termination and subsequent division. This cell cycle has not been looked at here, or do I miss something?

When we stated that FtsZ treadmilling dynamics are cell cycle ‘regulated’, we were referring to the change in the percentage of treadmilling filaments over time. The distribution of treadmilling speeds also shows more variation in nascent Z-rings (see Fig. 1h). **We have now made this more explicit in the Results section (L140-141):**

“The results show that the fraction of treadmilling FtsZ filaments and their speeds are division stage regulated.”

The reviewer raises a fair point that our investigation does not quite look at FtsZ treadmilling during all stages of the ‘cell cycle’, as these stages are typically defined by DNA replication initiation and termination as well as cytokinesis. **We have now replaced this with ‘division process’ or ‘division stages’ throughout the manuscript as appropriate.**

7. The authors use the GTPase inhibitor PC190723 to stop FtsZ treadmilling. Control experiments show that the effect on treadmilling is immediate, however, in the original publication (Haydon et al., 2008) addition of PC190723 leads to a dispersed, punctate localization of FtsZ. Benzamides were shown to enhance FtsZ bundling and therefore I am not sure if PC addition is ideally suited to control treadmilling behavior.

The reviewer is right to point out that PC19 treatment causes multiple observable effects on FtsZ including lower GTPase activity, dispersed localization, and enhanced bundling. However, these are all downstream effects of PC19 consistent with treadmilling inhibition. From the literature on the mechanism, PC19 binds a cleft in FtsZ distant from the GTP binding site (Haydon et al. 2008, Tan et al. 2012, Matsui et al. 2012, Elsen et al. 2012, Adams et al. 2016) – and only in a small subset of Gram-positive species related to *B. subtilis* – that locks FtsZ into an ‘open’ conformation (Wagstaff et al. 2017, Artola et al. 2017). This stabilizes the polymeric form of FtsZ (Haydon et al. 2008, Andreu et

al. 2010) and therefore prevents treadmilling (Bisson-Filho et al. 2017). GTPase activity drops (to ~25% that of untreated FtsZ, using excess compound) likely because nucleotide exchange is impaired when there is less subunit turnover. **We have now more accurately described the mechanism of PC19 in the Results section (L161-164):**

“The antibiotic PC190723 (PC19) arrests FtsZ treadmilling in *B. subtilis* and related species by supporting the 'open' conformation of FtsZ and stabilizing protofilaments^{1,18-21} without fully abolishing subunit turnover or GTPase activity²².”

The dispersed, punctate localization of FtsZ mentioned by the reviewer is also clearly seen in our experiments immediately following PC19 treatment (See Supp. Videos 10-13, 15). These foci likely appear when FtsZ monomers from the cytoplasmic pool form filaments at random locations throughout the cell. The immobile filaments still exhibit subunit turnover (Adams et al. 2011).

For further discussion of PC19 mode of action and additional chemical perturbations we have now performed, see our below response to Reviewer #3 comment 8.

This is in particular important since in E. coli effects are described were PC might have targets other than FtsZ polymerization.

To our knowledge, non-FtsZ targets of PC19 have to date only been described in *E. coli* cells with either genetic or chemical perturbations that “increase outer membrane permeability, conferring increased sensitivity to compounds such as bile acids, SDS, and vancomycin” (Khare et al. 2019). This study found that spontaneous mutations conferring resistance to PC19 in membrane-permeabilized strains of *E. coli* map to nine non-division proteins, and the authors concluded that the effects of PC19 in *E. coli* are FtsZ-independent. However, this is somewhat unsurprising given that the binding pocket on FtsZ that PC19 binds only exists in certain Gram-positive species related to *B. subtilis*, and not in *E. coli*. In fact, PC19 does not affect *E. coli* FtsZ in vitro even up to a stoichiometry of 2 PC19 to 1 FtsZ (Andreu et al. 2010).

In contrast to the recent *E. coli* study, the original work describing PC19 found that mutations conferring resistance to PC19 in species that are naturally sensitive to it (e.g. *S. aureus* and *B. subtilis*) all mapped to FtsZ – specifically residues coinciding with residues of tubulin that form the binding site for Taxol (Haydon et al. 2008, Adams et al. 2016). Crystal structures of PC19 bound to *S. aureus* FtsZ have shown this specific binding pocket for the compound (Tan et al. 2012, Matsui et al. 2012, Elsen et al. 2012). Furthermore, the lethal effects of a related benzamide (3-MBA) in *B. subtilis* can be suppressed solely by mutations in FtsZ (Ohashi et al. 1999). Since there appears to be no evidence in the literature for an FtsZ-independent mechanism for PC19 in *B. subtilis* – yet strong evidence to the contrary – we believe the effects observed in our experiments result solely from the effect of PC19 on FtsZ. **We have now clarified in the Results section that PC19 most likely arrests FtsZ treadmilling in B. subtilis-related species specifically (L161-164):**

“The antibiotic PC190723 (PC19) arrests FtsZ treadmilling in *B. subtilis* and related species by supporting the 'open' conformation of FtsZ and stabilizing protofilaments^{1,18-21} without fully abolishing subunit turnover or GTPase activity²².”

I therefore like the use of the FtsZ treadmilling mutants such as D213A (I assume this is the equivalent mutant to D212G described in E. coli (Yang et al., 2017). Use of this mutant should be explained better in the text to make this point clear.

We thank the reviewer for their positive comments regarding the D213A experiments and have now expanded upon these experiments to complement our chemical perturbations. Our original manuscript showed (as also described in Bisson-Filho et al. 2017) that FtsZ(D213A) slowed FtsZ treadmilling speed and consequently slowed septal constriction rate, but did not show the effect on Z-ring condensation. We have now provided data examining the effect of FtsZ(D213A) expression on Z-ring condensation as a complement to the chemical perturbations. Expression of the mutant protein slowed FtsZ treadmilling speed and also led to significant effect on Z-ring condensation: fewer rings successfully condensed prior to constriction initiation, and even constricting rings were less condensed (i.e. thicker) than in wild-type. **We now have more clearly described the original D213A experiments. We have included the new D213A results as Supp. Fig. 16, Supp. Video 17, Supp. Fig. 17, and have described them in the Results section (L237-252):**

“To further investigate the role of FtsZ treadmilling in Z-ring condensation, we examined the effect of an FtsZ mutant that is deficient in GTP hydrolysis. The mutant FtsZ(D212A) in *E. coli* was shown to have greatly reduced GTPase activity^{28,29}, and exogenous expression of the equivalent FtsZ(D213A) mutant in *B. subtilis* reduces FtsZ treadmilling speed¹ (Supplementary Figure 16). We constructed a merodiploid strain expressing FtsZ-GFP from the native locus and FtsZ(D213A) from a secondary locus under an IPTG-inducible promoter (SH131, Methods). For comparison to wild-type (Figure 1) we imaged this strain under slow growth conditions (poor media, 30°C) with an induction level (10 μM IPTG) that supported growth in liquid culture (Supplementary Figure 3) and cell division (Supplementary Video 17). Condensation of nascent Z-rings was significantly perturbed under these conditions: 54% (N=74) of nascent Z-rings failed to fully condense prior to constriction initiation, and even throughout constriction Z-rings remained 65 nm (95% CI [63,67]) thicker than wild-type (Supplementary Figure 17), consistent with previous measurements in *E. coli*³⁰. This partially deleterious effect on Z-ring condensation is consistent with partial inhibition of FtsZ treadmilling caused by expression of the GTPase-deficient mutant. Interestingly, many incompletely-condensed Z-rings were still able to constrict, implying that cells can tolerate some degree of error in the condensation step.”

8. Suppl. Figure 1: Indicate how much lysate (total protein) was loaded.

We have now provided this in the caption to Supp. Fig. 1.

Ideally provide internal control and show blots for all strains used in this study that harbor FP fusions.

We have now replaced Supp. Fig. 1 with Western blots for all relevant strains harbouring FP fusions (SH130, ME7, SH212, bWM4). For strains expressing FP fusions from inducible promoters (SH212 and bWM4), cultures were grown with the concentration of inducer used in experiments prior to blotting.

Reviewer 2 (Remarks to the Author):

Overall this is a very nice manuscript that uses an elegant microfluidic imaging platform to visualize the motion of FtsZ filaments in bacillus subtilis. The platform enables quantitative analysis of FtsZ motion, septal constriction, and response of this process to chemical perturbation to provide some basic insights into the role of FtsZ treadmilling in cell division.

The videos in the supplement are very impressive, and the image processing, analysis and extraction of quantitative information was adequately described, reasonably justified, and transparent. The methods were cited appropriately when used in previous work, allowing reasonable follow up by the reader.

Thank you for the positive comments. All specific comments are addressed below.

Some questions/concerns that I would like addressed...

1. In the initial description of the 3 "distinct" phases of z-ring stages (nascent, mature, constricting, I was not fully convinced based on figure 1B that these phases were quite distinct. Visually there is a marked change in the slope of the curves in 1B when cells begin constricting, but the use of a cutoff of 400nm to distinguish between nascent and mature seems weak. 1C is colored to show 3 clusters of cells, but it is not clear how distinct these clusters really are.

In all, the use of these classifications as descriptive and speculative seems fine for the sake of discussion. However, a description of these as "distinct" phases begs for stronger justification. Also please consider the next comment below and how it might impact distinction between stages etc.

The assignment of distinct nascent and condensed pre-constricted cell division states was based upon the clear but qualitative observation of a rapid condensation in axial time series of Z-ring thickness during early Z-ring assembly (Fig. 1b). However, previously classification was based on a simple axial thickness threshold. **We have now replaced the simplistic axial-thickness-based classification with direct step finding analysis of the axial thickness time traces to segregate decondensed (nascent) and condensed (mature) Z-rings (Fig. 1, Supp. Fig. 4, Methods, Supplementary Note 3).** Step detection supports our conclusion that Z-rings condense rapidly and stochastically on average 10 minutes before constriction initiation. Fig. 1 has been updated to incorporate the new step-finding state assignment method. Additional results of step detection are shown in Supp. Fig. 4 together with new analysis of timing of Z-ring condensation (Supp. Fig. 4b).

2. Throughout the manuscript overall thickness and diameter measurements are used to develop scatter plots and cluster cells. This would benefit from a clearer picture of variation in cell size and diameter for the normal population of cells. For example the cells are described as "normal" versus "mildly elongated" at different temperatures (82-83). What was the actual difference in length and how did this vary for the population?

The distributions of lengths for SH130 cells (the strain used in most experiments with corresponding scatter plots) under different growth conditions can be found in Supp. Fig. 2. For diameters, see below response.

Can diameters, be expressed as % change in diameter for each cell? Were you able to capture the actual diameter or starting diameter of a cell versus the z-ring diameter. Expressing normalized % diameter would likely reduce some of noise in the scatter plots and collapse some of the groupings. (i.e. a cell that appears to have a smaller diameter ring may just be a smaller cell and not really be further along in the ring formation process.)

Alternatively, providing a description of the size distribution for the population of cells used and showing that the population does not vary in diameter or length significantly compared to the variation in diameter seen in the z-rings would provide adequate justification to not perform a normalization. (i.e. if everything is divided by the same number...no need to normalize)

We have now gone through all time-lapses for chemical perturbation experiments and measured the initial (i.e. maximal) diameters of each Z-ring where possible so as to normalize each data point in the scatter plots by its own initial diameter. All scatter plots for chemical perturbation data that had used “Diameter (nm)” as an axis have now been replaced with scatter plots using “Relative Diameter” as an axis. As the reviewer suggested, this has reduced the noise in the scatter plots significantly.

3. Please discuss the use of agarose versus PDMS in different stages more clearly. Describe why the change was made and how the use of the hydrogel versus the PDMS might impact transport of nutrients, cell health and response.

We primarily chose to use PDMS rather than agarose for the microfluidic VerCINI method due to the difficulty of producing an agarose pad thin enough to image through. In the original VerCINI method the cells are adjacent to the coverslip while the agarose pad is above them, and so the thickness of the pad is not relevant for imaging. In contrast, in microfluidic VerCINI the pad is adjacent to the coverslip with cells above, and so this poses a unique challenge for imaging. As noted in the Methods section, even the 50 μm layer of PDMS was already thick enough to necessitate a separate objective with high working distance. **We have now included brief mention of this rationale in the Results section (L171-172):**

“PDMS was used rather than agarose in order to produce a thin enough layer ($\sim 50 \mu\text{m}$) to image through.”

The reviewer raises an interesting point about the response of cells to growth in agarose vs. PDMS microholes. The transport of nutrients is very likely different in the two cases: by diffusion through the hydrogel for agarose microholes or continual flow for PDMS microholes. Nevertheless, we have observed that growth and division of cells in both agarose and PDMS microholes continues for at least 1 hour – well within the duration of our measurements. This is consistent with experiments and modelling showing that the physiology of *E. coli* cells is not significantly affected by growth in short ($\leq 20 \mu\text{m}$) PDMS channels (Yang et al. 2018), and that specifically growth of *B. subtilis* cells is not impaired in channels wider than $\sim 0.8 \mu\text{m}$ (Mannik et al. 2009). For comparison, our microholes (agarose and PDMS) are $< 8 \mu\text{m}$ deep and $> 1.0 \mu\text{m}$ wide. **We have now included mention of the effect on cell health in the Results section (L172-175):**

“Cells continued to grow and divide, consistent with previous observations that growth of *B. subtilis* cells in sufficiently wide ($> 0.8 \mu\text{m}$) PDMS channels is not impaired²³, and models demonstrating that nutrient transport does not significantly limit growth in short ($< 20 \mu\text{m}$) channels²⁴.”

4. You conclude that the motion of FtsZ leads to more collisions and condensations. This is fairly

obvious or intuitive, but it is nice to visualize. However, there seems to be a lack of discussion as to why this occurs in the center of the cells. Is there something in the curvature of the cell ends that drives the FtsZ to the center of the cell, causing a split? Is there molecular crowding at the ends of the cells that drive FtsZ to the center? Is this something that could be modeled and addressed more deeply in discussion? By itself, the finding that motion leads to more collisions, leads to condensation, seems like an obvious statement and rather uninteresting. What drives this to occur at the center of the cell and allow an even split?

There are several factors well established in the literature causing FtsZ to preferentially polymerize near the centres of cells in *B. subtilis*. The Min system induces filament disassembly near poles, while the Noc system induces disassembly over the nucleoid. These two systems ensure that the only region available for appreciable filament formation is the centre.

In contrast to these well-established factors, our discovery of filament collisions has not been reported before in the literature. FtsZ filaments were only shown to be motile *in vivo* recently (Bisson-Filho et al. 2017, Yang et al. 2017), and so the discovery that moving filaments collide and aggregate into a dense Z-ring demonstrates a novel function for FtsZ.

Our finding that FtsZ motion leads to Z-ring condensation is not immediately obvious. One previous model for Z-ring condensation was that an extended, helical FtsZ structure slides together laterally as a spring (see model in Monahan et al. 2009 and discussion in Erickson et al. 2010). **We have now contrasted our results with this previous model in the Discussion section (L347-351):**

“Our results reveal an additional key factor in assembling the Z-ring: FtsZ filament treadmilling drives filament interactions by enabling FtsZ filaments to rapidly search the mid-cell surface circumferentially and efficiently encounter one another. This finding contrasts with previous models of Z-ring condensation involving lateral sliding of filaments^{13,14.}”

Overall this is a very nice manuscript and highlights a nice method for visualizing molecular processes in bacterial cells. I recommend publication of the manuscript once the above concerns/comments are addressed.

*Regards,
Scott T. Retterer*

Reviewer 3 (Remarks to the Author):

*In the manuscript “FtsZ treadmilling is essential for Z-ring condensation and septal constriction initiation in bacterial cell division”, Whitley et al. use novel and existing microscopy techniques to resolve differences in FtsZ dynamics throughout the stages of the cell cycle in *Bacillus subtilis*. The authors determined that *B. subtilis* constriction can occur independently of FtsZ polymer dynamics after constriction activation. This observation was made possible by creative use of “micropillars” to vertically position and image *B. subtilis* cells both in agarose “microwells” and in a microfluidics device under flow. Positioning cells vertically, de-noising the images by mathematically enhancing the signal-to-noise ratio, and using more accurate models for calculating constriction rate have all contributed to improved resolution of how FtsZ dynamics and treadmilling contribute to constriction. These methods, combined with the use of the established FtsZ inhibitor, PC190723 (PC19), demonstrate that FtsZ treadmilling is halted in cells within a few seconds of treatment with PC19. Their approach allows the authors to classify constriction timing and determine that when FtsZ dynamics are halted with PC19, cells will only constrict if constriction has already initiated. Lastly, the authors observe that FtsZ dynamics are independent of the temperature or richness of the medium, suggesting that the rate-limiting step of cell division in *B. subtilis* is growth rate rather than FtsZ dynamics.*

*The major strength of this study is the novel microscopy techniques employed, which arguably provide the best resolution of FtsZ dynamics and constriction yet presented. Despite the beautiful use of these techniques, the novel mechanistic insights that this manuscript presents are bit more limited. The finding that FtsZ dynamics are not required for constriction in *B. subtilis* after initiation of constriction echoes a similar finding in *Staphylococcus aureus* (Monteiro et al 2018 Nature). The major difference between the two may be that dynamic FtsZ still contributes to an optimal constriction rate in *B. subtilis*, but the contribution of FtsZ dynamics to constriction rate in *B. subtilis* was also previously established (Bisson-Filho et al 2017 Science).*

We thank the reviewer for their constructive comments on our manuscript.

The reviewer correctly describes that our results are in agreement with those of Monteiro et al. 2018 and Bisson-Filho et al. 2017, but, respectfully, we do not believe they have fully appreciated the novelty of the mechanistic insights from our work. In addition to our discovery of the role of FtsZ treadmilling in Z-ring assembly (discussed below), our results reconcile contradictory models for the role of FtsZ in septal constriction and therefore provide a unified mechanistic model for FtsZ treadmilling in cell division across key Gram-positive model organisms.

The Bisson-Filho et al. 2017 paper demonstrated (in *B. subtilis*) that FtsZ treadmilling speed affected septal constriction rate, and from this presented a model whereby FtsZ treadmilling was the central, essential factor governing/guiding the full constriction process. Based on this model, one would expect (as we did) that arresting FtsZ treadmilling would consequently arrest septal constriction. In contrast, the Monteiro et al. 2018 paper showed (in *S. aureus*) that FtsZ treadmilling is dispensable for constriction after an initiation step, and presented a model whereby peptidoglycan synthases – not FtsZ – are the central factors governing the constriction process. How can FtsZ treadmilling be dispensable for constriction while at the same time be the rate-determining factor? Our study reconciles this apparent contradiction: FtsZ treadmilling is dispensable after initiating constriction, *but it is still a significant factor determining the rate.*

Furthermore, we show the effect of FtsZ treadmilling speed on cell constriction is much more modest than we previously thought, and in fact overall cell growth rate has a much larger impact on

septal constriction rate than FtsZ treadmilling speed does. This is a major departure from our FtsZ-centric model of constriction in Bisson-Filho et al. 2017.

To address these concerns, we have extended the discussion of the significance of our findings in the context of current literature. We have now better described how our findings contrast with previous studies in the Results section (L297-301):

“These results show that although FtsZ treadmilling is not required for septal PG synthesis after constriction initiation, treadmilling promotes efficient constriction by increasing septal constriction rate. This therefore reconciles apparently contradictory models previously proposed for the role of FtsZ treadmilling in septal constriction^{1,3}.”

and in the Discussion section (L367-370):

“We conclude that FtsZ treadmilling plays only a secondary role in active constriction, while its critical roles occur prior to constriction initiation. This contrasts with the previous model of *B. subtilis* FtsZ as an obligatory guide and rate-determining factor for constriction¹.”

The conclusion that FtsZ treadmilling contributes to “Z-ring condensation” is not as well-developed here, and it has long been established that FtsZ dynamics are important for Z-ring formation and structure at midcell. Overall, the paper is well-written, the microscopy is really impressive and allows the most comprehensive view of B. subtilis Z-ring dynamics and function yet presented, but the novelty of the mechanistic insights may be more limited.

We agree with the reviewer that the conclusions about the role of FtsZ treadmilling in Z-ring condensation could be strengthened. We have now better developed this conclusion by providing results using a merodiploid strain expressing FtsZ-GFP from the native locus and the GTPase-deficient FtsZ(D213A) at a secondary locus from an inducible promoter so that we could investigate the effect of the mutant protein on Z-ring condensation. Expression of the mutant slowed FtsZ treadmilling speed (as shown previously in Bisson-Filho et al. 2017) and also led to significant effect on Z-ring condensation: fewer rings successfully condensed prior to constriction initiation, and even constricting rings were less condensed (i.e. thicker) than in wild-type. **We have provided these results in Supp. Video 17, Supp. Fig. 17, and have described them in the Results section (L237-252; see response to Reviewer #1 comment 7).**

However, respectfully, we also do not think the reviewer has fully appreciated the novelty of these results. We agree that it was already known that FtsZ dynamics, in the very basic sense of turnover of FtsZ filaments at mid-cell, is important for assembling the Z-ring prior to constriction initiation. However, until now it was absolutely not clear why this was the case. *Our important novel finding here is to provide biophysical mechanism for how FtsZ dynamics drive Z-ring assembly, by showing that it is FtsZ treadmilling motility that drives Z-ring condensation.* **We have addressed this concern regards novelty of the findings relating to Z-ring condensation by adding detailed discussion of this point in the Discussion section (L345-354):**

“Chemical arrest of treadmilling completely abolished FtsZ ring condensation. Attractive lateral interactions between FtsZ filaments, via the FtsZ C-terminal-linker and cytoplasmic FtsZ-associated proteins are known to mediate FtsZ bundling and Z-ring condensation^{34–38}. Our results reveal an additional key factor in assembling the Z-ring: FtsZ filament treadmilling drives filament interactions by enabling FtsZ filaments to rapidly search the mid-cell surface circumferentially and efficiently encounter one another. This finding contrasts with previous models of Z-ring condensation involving lateral sliding of filaments^{13,14}. We conclude that FtsZ treadmilling—likely along with FtsZ lateral interactions—drives aggregation of FtsZ filaments into a condensed Z-ring. This is remarkably

reminiscent of treadmilling-driven partitioning of FtsZ filaments into higher-order structures previously observed *in vitro*^{9,39}.”

1. Throughout, the authors use PC19 to impact/prevent FtsZ dynamics. However, it's important to note that PC19 does not specifically stop treadmilling, but impacts FtsZ GTPase activity and filament structure (Haydon et al 2008 Science, Adams et al 2011 Mol Micro). Use of PC19 therefore does not allow one to specifically comment on the role of treadmilling, but of FtsZ dynamics/structure overall. This should be mentioned and elaborated upon in the context of the findings of this paper that involve PC19 use.

The reviewer is right to point out that PC19 does not only stop treadmilling, but also impacts GTPase activity and filament structure. However, these are downstream effects of PC19 consistent with treadmilling inhibition (*the mechanism of PC19 is further discussed above in response to Reviewer #1 comment 7*). In addition to the above discussion, we note that FtsZ retains ~25% GTPase activity even with excess PC19 (Haydon et al. 2008). Furthermore, in cells treated with 8j (a relative of PC19) the subunits in immobile FtsZ foci still turn over slowly (Adams et al. 2011), which suggests filaments are still dynamic despite not treadmilling. We note also that in the work of Adams et al 2011, long extended FtsZ filaments were only induced by 8j treatment at sub-MIC levels. Super-MIC treatment with 8j similar to the conditions of our PC19 experiments only induced formation of small FtsZ foci, consistent with our observations.

So, while we cannot strictly rule out the possibility that our observations result from a downstream effect of PC19 and while we agree that it is important to discuss these limitations, we believe the most parsimonious explanation for our results is that treadmilling arrest specifically is responsible. **We have added discussion of the mode of action and limitations of our PC19 experiments to the Results section (L219-225):**

“It is worth noting that PC19 also affects FtsZ GTPase activity and filament structure¹⁸. These are likely downstream effects of the primary mode of action, which is to stabilize the polymeric state of the protein. We note that FtsZ retains ~25% GTPase activity even with excess PC19¹⁸. Furthermore, in cells treated with 8j (a relative of PC19) the subunits in immobile FtsZ foci still turn over slowly²², which suggests filaments are still dynamic despite not treadmilling. So, while we cannot strictly rule out the possibility that our observations result from a downstream effect of PC19, we believe the most parsimonious explanation for our results is that the arrest of treadmilling specifically is responsible.”

We have also performed additional experiments to supplement our original measurements of cell division in an FtsZ GTPase mutant, including quantitative analysis of perturbations to FtsZ-ring condensation in this strain, which support our conclusion that FtsZ treadmilling arrest is primarily responsible for the perturbations to cell division reported in this manuscript (Supp. Fig. 16, Supp. Video 17, Supp. Fig. 17, main text L237-252, see also Reviewer #1 comment 7, Reviewer #3 comment 8).

2. Title: The paper does not address “bacterial cell division”, it addresses B. subtilis cell division. This is important to clarify since there are well-documented differences in Z-ring dynamics and their relationship to cell wall metabolism between B. subtilis and other bacterial species (e.g. E. coli and S. pneumoniae).

We agree with the reviewer on this point, and **have now made this change to the title.**

3. Throughout, the authors allude to the “movement” or “motility” of treadmilling FtsZ filaments. Since treadmilling is sometimes a difficult concept for non-experts to understand, it’s important to be very clear that it is apparent motion/motility/movement and that single molecules of FtsZ do not move.

We have now changed terms relating to “motion” or “motility” throughout the manuscript where appropriate to avoid confusion.

4. Line 118 and Figure 1eii: It is hard to determine when two clusters collide and become temporarily arrested. Can you highlight this in the figure?

We have now added small arrows to kymographs to highlight clear filament interaction events (Figure 1, one example highlighted, Figure S7, multiple examples highlighted).

5. It is surprising that nascent rings appear to have reduced treadmilling compared to mature and late rings. There are numerous studies in a range of bacteria showing that Z-ring localization is highly dynamic early in the cell cycle and becomes more stable later. This would imply that FtsZ filaments are at least as dynamic in nascent rings as in mature rings, and perhaps more so. Mechanistically, it’s also difficult to imagine how FtsZ filaments would be more stable in nascent rings. Could this be an artifact of the difference in density of clusters in nascent vs mature rings? That is, as rings become denser, it becomes much more difficult to pick out individual FtsZ clusters. Perhaps non-moving clusters are obscured and under-counted in mature and constricting rings? If the authors don’t think this is a technical artifact, some further explanation is required for how/why nascent rings might not treadmill. Another possibility is that, in fact, the nascent FtsZ structures are still very dynamic, but do not undergo treadmilling (i.e. if you performed FRAP on nascent rings, they would show similar dynamics of recovery as mature rings).

We have excluded the possibility that immobile FtsZ filaments are a technical artefact of undercounting in mature/ constricting rings by performing simulations to test this hypothesis (Supp. Fig. 10, Methods)

Regards comments comparing dynamics of FtsZ filaments in nascent and mature Z-rings. To clarify, we did not aim to suggest that our data showed that filaments in nascent FtsZ rings were less dynamic in the sense of having slower turnover/ longer binding time at midcell. Rather, we found that a large population of FtsZ filaments in nascent FtsZ rings are immobile, i.e. do not move processively, but this does not mean that subunit turnover within filaments is affected.

We believe the problem here is our unclear use of both the terms ‘static’ and ‘immobile’ FtsZ filament to describe filaments that do not undergo treadmilling, which leads to confusion about turnover of FtsZ filaments in nascent Z-rings. **We have now changed to consistently using the term *immobile FtsZ filament*.**

As the reviewer suggested, we showed FtsZ filaments in nascent Z-rings have similar binding lifetimes to mature/ constricting FtsZ filaments (Supp. Fig. 7c (revised: 9c)). If subunit turnover were slower in immobile filaments, we would expect that the filament lifetime would increase. However, we see that filament lifetime remains mostly constant for both motile and immobile filaments across division stages, suggesting that subunit turnover within filaments is not significantly different. **We have now included data to show that motile and immobile filaments have similar lifetimes (Supp. Fig. 11).**

Regarding why nascent Z-rings might not treadmill: We note here Rev #3's other comments about lack of novelty and we wish to point out that the findings that FtsZ filaments form a mixed population of treadmilling/ non-treadmilling filaments, whose ratio is strongly division stage regulated, are unexpected and novel, with both Rev #1 and #3 commenting on this point and asking for extended discussion/ investigation of this surprising observation. **We have now added further discussion of the mechanistic basis for immobile FtsZ filaments and their turnover rate (L342-344 and see response to Rev #1 comment 3).**

Related, Line 122: "drop in the number of immobile FtsZ filaments" could just be a drop in the ability to detect immobile FtsZ filaments.

As noted above we have now excluded this hypothesis via simulation.

6. One of the major conclusions of the paper is that FtsZ treadmilling is required for Z-ring condensation, but the data presented to support this are fairly limited. In figure 3b and supplementary figure 8, how was "condensed" vs "not condensed" determined?

The reviewer raises a good point that the paper did not present much raw data or provide much description of the analysis used to produce the results shown in Supp. Fig. 8 (revised: 13).

Detailed automated measurements of Z-ring dynamics in wild type cells (Fig 1), now supplemented with additional step-detection quantification, established clearly that Z-rings undergo rapid stochastic condensation, i.e. transition from a sparse axially extended array of FtsZ filaments to a narrow dense Z-ring, approximately 10 minutes prior to constriction initiation.

In the microfluidics experiments (Fig 3) whether or not Z-rings condensed (or constricted) post-PC19 treatment was evaluated manually. We found that this was the most robust way to do the analysis due to the difficulty and inaccuracy of measuring Z-ring dimensions post-PC19 treatment in strains expressing labelled FtsZ. Since PC19 treatment causes bright fluorescent foci of FtsZ to appear randomly throughout the cell, tracking software fails to consistently follow most Z-rings post-treatment in the SH130 and SH212 strains (this is the reason we used a strain expressing mNeonGreen-PBP2B (ME7) to track rings for comparison of constriction rates pre- and post-treatment: PBP2B does not form bright foci randomly throughout the cell that would interfere with tracking). Not only this, but measurements of Z-ring thickness will not be accurate post-treatment due to bright foci in the vicinity of Z-rings (i.e. thickness will frequently be overestimated, potentially giving false 'non-condensed' results even if a ring actually condensed post-treatment).

Although automated quantification of axial Z-ring thickness is not currently possible in PC19 treated cells, we found that Z-ring condensation state was straightforward to detect manually, and therefore proceeded with manual Z-ring classification. **We have now provided example image sequences of nascent rings condensing or not condensing post-PC19 treatment to Supp. Fig. 8 (revised: 13), and have also provided a description and rationale of this manual analysis to the Methods (L652-657):**

"Evaluation of percentage Z-rings condensed and constricted post-treatment

Since PC19 treatment causes bright fluorescent foci to appear randomly throughout the cell in strains where FtsZ is labelled, tracking software fails to consistently follow most Z-rings post-treatment in the SH130 and SH212 strains. Furthermore, measurements of Z-ring thickness and diameter were

inaccurate due to the proximity of fluorescent foci. Since it was typically straightforward to observe, whether Z-rings condensed and/or constricted post-PC19 treatment was evaluated manually.”

Moreover, we have now provided more data to strengthen this conclusion by using the FtsZ(D213A) mutant (where automated image analysis of condensation was used) Methods, Supp. Fig. 16, Supp. Video 17, Supp. Fig. 17, Results section L237-252. As described above, reducing treadmilling through induction of the mutant significantly impaired Z-ring condensation while still allowing cell division: fewer Z-rings fully condensed prior to constriction initiation, and even actively constricting rings were less condensed than those in wild-type.

7. Line 117: why is it surprising to observe clusters of FtsZ colliding and fusing/aggregating?

To the best of our knowledge, up until now, interaction and/or aggregation of motile FtsZ filaments has never previously been directly observed either *in vitro* or *in vivo*. We agree that in principle some degree of FtsZ filament interaction is expected based on strong evidence for indirect and direct FtsZ-FtsZ lateral interactions. However, what is very surprising to us is that this interaction is strong enough to frequently cause temporary arrest or direction change of one or both filaments (blue lines, Fig. 1e and Supp. Fig. 5 (revised: 7)). In support of this, we note that a previous *in vitro* study showed that transient filament interactions via ZapA stabilize FtsZ treadmilling without perturbing FtsZ filament speed (Caldas et al. *Nat Comm* 2019). Our observations show evidence for stronger filament interactions with more perturbative effect on FtsZ filament speed than was previously expected.

Line 120: it may not be “treadmilling” of FtsZ that specifically drives filament encounters, but FtsZ dynamics in general.

We state in L125-126 that “these data show that FtsZ treadmilling can drive filament aggregation by promoting the filament encounters required for lateral interactions.” The data to which we are referring is the direct observation of motile filaments traversing the cell circumference, encountering other filaments, and aggregating (Fig. 1c, Supp. Fig. 5 (revised: 7), blue lines). If these FtsZ filaments were not motile they would not encounter each other, so we think that our statement stands. We agree that this does not exclude the possibility that dynamic binding/ unbinding of immobile FtsZ filaments from mid-cell also contributes to FtsZ filament aggregation.

8. It would be lend confidence and specificity to the conclusions if the authors were able to complement the PC19 treatment with other ways of perturbing the Z-ring and its dynamics. For example, how is PBP2B intensity impacted in strains producing FtsZ GTPase mutants that have slower treadmilling? What happens to constriction if Z-rings are disassembled (e.g. with expression of MciZ)?

We thank the reviewer for this helpful suggestion. **Firstly, we have performed additional experiments to supplement our original measurements of cell division in an FtsZ GTPase mutant, including quantitative analysis of perturbations to FtsZ-ring condensation in this strain (Supp. Fig. 16, Supp. Video 17, Supp. Fig. 17, main text L237-252, see also Reviewer #1 comment 7).** These measurements support our conclusion that FtsZ treadmilling drives Z-ring condensation and promotes septal constriction.

Secondly, based on the reviewer's suggestion, we have now also performed experiments using additional chemical perturbations to supplement our PC19 results. We describe two additional experiments and relevant results of a new preprint:

- MciZ:** This peptide inhibits Z-ring formation, possibly by acting as a 'cap' on FtsZ filaments (Bisson-Filho et al. 2015). MciZ also prevents filament assembly (Araújo-Bazán et al. 2019) and leads to a diffuse cytoplasmic signal (Bisson-Filho et al. 2015). **We therefore repeated the experiments of Fig. 3a-b with exogenous addition of *B. subtilis* MciZ (20 μ M).** However, we do not see appreciable filament disassembly for \sim 20 min post-addition, which is long enough for most Z-rings identified at $t=0$ to fully condense and constrict. The slow timescale of action of MciZ prevents us from separately determining the effect of MciZ during different stages of cells division, and so prevents us from drawing useful conclusions in the context of this manuscript. Therefore we do not propose to include these results in the manuscript. **However, for completeness and based on the reviewer's suggestion we are providing the results to the reviewers as a figure (below) and video (uploaded with the revised manuscript).**

Reviewer MciZ video is included as manuscript item "Video - Video of MciZ perturbation": Video shows time-lapse of response of strain SH130 expressing FtsZ-GFP to treatment with 20 μ M *B. subtilis* MciZ (Peptide Specialty Laboratories GmbH). Video time stamp marks time post-MciZ treatment in minutes.

Reviewer MciZ Figure:

Figure MciZ: Septal completion assay post-MciZ treatment. (a) Scatter plot of Z-ring diameters (normalized by maximal diameter for each Z-ring) and thicknesses for all FtsZ-GFP cells at $t=0$ min showing whether they continued constricting (blue) or not (red) following treatment with 20 μ M *B. subtilis* MciZ (Peptide Specialty Laboratories GmbH). (b) Percentage of cells that continued constricting classified by division stage. Nascent: thickness >400 nm, mature: thickness <400 nm, relative diameter >0.9 , constricting: thickness <400 nm, relative diameter <0.9 . (c) Percentage of cells in the nascent Z-ring stage that condensed post-MciZ compared to the percentage that condensed post-PC19. Numbers above stacked bars indicate number of cells.

- PC58538:** This compound (a.k.a. PC58), first described by Stokes et al. 2005, has a separate mode of action (and separate binding site) from PC19: Rather than stabilizing FtsZ filaments,

PC58 prevents filament assembly. Importantly, because of this the compound does not promote bundling or the formation of dispersed foci after treatment like PC19 does; it actually has the opposite effect – causing a uniform, diffuse cytoplasmic FtsZ-GFP signal (Stokes et al. 2005). We have now repeated the experiments of Fig. 3a-b with this compound. When treated with PC58, nascent Z-rings typically fail to condense. However septa which have initiated constriction usually continue. We conclude that the effect on cell division initiation of FtsZ treadmilling inhibition by PC19 is similar to, and as severe as, preventing FtsZ filament assembly at mid-cell by PC58, emphasizing the critical role of FtsZ treadmilling in cell division initiation. **We have now included these results as a Supplementary Figure (Supp. Fig. 15) and Video (Supp. Video 16), and have described it in the Results section as a second chemical perturbation to FtsZ treadmilling that is complementary to PC19 (L226-236):**

“We considered whether these effects were specific to PC19 by repeating the assay with the non-benzamide FtsZ inhibitor PC58538 (PC58). This compound inhibits FtsZ by a separate mode of action from PC19: Rather than stabilizing protofilaments, it prevents filament assembly²⁶. As with PC19, PC58 prevented 84% (N=50) of nascent Z-rings from condensing into mature Z-rings and 90% (N=50) from constricting, but prevented 60% (N=52) of mature and only 6% (N=33) of constricting Z-rings from constricting (Supplementary Figure 15, Supplementary Video 16).”

- **ADEP:** ADEP compounds inhibit division by activating ClpP to degrade FtsZ monomers (Brötz-Oesterhelt et al. 2005, Silber et al. 2019). A recent pre-print showed that treating *B. subtilis* cells with ADEP results in nascent Z-rings failing to condense, and mature Z-rings failing to constrict unless PBP2B is already present (Silber et al. 2020 *bioRxiv*), similar to the effect of PC58 treatment described above. **We have now referenced this pre-print in the Results section (L231-232).**

9. Line 221: since the role of FtsZ dynamics in formation of a focused midcell ring is well-established, it's not clear how "novel" the role of Z-ring dynamics in "condensing" the Z-ring is. PC19 was previously shown to disrupt and extend Z-ring structures, e.g. in Adams et al 2011 Mol Micro.

As the reviewer rightly points out, Adams and coworkers previously showed that inhibiting FtsZ filament depolymerization via benzamide antibiotics causes inhibition of FtsZ ring assembly, mislocalization of FtsZ into small foci above the MIC, and formation of extended FtsZ filaments below the MIC. These observations are made as endpoint assays after substantial treatment time, and are thus unable to provide insight into how benzamide treatment perturbs FtsZ ring assembly, which happens rapidly and requires time resolved measurements.

We agree that it was already known that FtsZ dynamics, in the very basic sense of turnover of FtsZ filaments at mid-cell, is important for assembling the Z-ring prior to constriction initiation. Until now it was not clear why this was the case. As noted above, *our novel finding here is to provide biophysical mechanism for how FtsZ dynamics drive Z-ring assembly, by showing that FtsZ treadmilling motility specifically drives a rapid aggregation transition between the dispersed and condensed FtsZ ring states.*

10. Line 94: It would be helpful to state the classification criteria (listed on lines 328-331 in the supplement) for the different ring states in the main texts for ease of reference and clarity.

Since our classification criteria for decondensed/condensed rings has now changed and is more complicated than a simple threshold (see response to Reviewer #3 comment 6), we believe it is most appropriate to leave this description to the Methods section and Supplementary Note 3. **We have, however, clarified that the classification is “based on Z-ring diameter and axial thickness” in the main text (L93-94).**

11. Line 111: More accurate to say figure 1e-h

We have made this change.

12. Line122-126: Data is in Figure 1h so that should be referenced.

We have now referenced Figure 1h.

13. Lines 131-132: The data in figure 1 do not show that “FtsZ treadmilling [promotes] Z-ring condensation prior to septal constriction initiation”.

Thank you for pointing this out. We have removed this phrase.

14. Line 218-220: It’s not clear what the authors mean by “FtsZ treadmilling [guides] septal synthases”. Directs their dynamic localization? Regulates their activity?

We have now clarified that we are referring to the processive motion of septal synthases (L276).

15. Line 283: This study does not investigate FtsZ lateral interactions at all, so conclusions about lateral interactions should be avoided.

We have now rephrased this to clarify that our discussion of FtsZ lateral interactions is speculative, and not a conclusion of this study (L351-352):

“We conclude that FtsZ treadmilling likely along with FtsZ lateral interactions drives aggregation of FtsZ filaments into a condensed Z-ring.”

16. Supplemental Line 21: “... at indicated temperature.” Add ‘the’

We have made this change.

17. Supplemental figure 6 appears to suggest there is a larger fraction of nascent rings measured based on the VerCINI method than what is measured based on the horizontal cell method. Do they have any comment on this?

We cannot say for sure, but is likely this is because VerCINI microhole arrays tend to preferentially trap cells that are shorter, i.e. earlier in the cell cycle, leading to a mild enrichment of these cells.

18. Supplementary figure 8: Is it significant that HT-PBP2 expressing cells are more likely to have condensed rings during PC19 treatment? Could this be the tag contributing to aggregation/an increase in density during division? Any explanation?

The SH212 strain (GFP-FtsZ, HaloTag-PBP2B) has significantly lower signal-to-noise ratio than the SH130 (FtsZ-GFP) strain due to lower expression of labelled FtsZ. This is due to the fact that labelled FtsZ is the sole source of FtsZ in SH130 (See Supp. Fig. 1) while it is present along with unlabelled FtsZ in the SH212 strain.

Due to the poorer SNR of this strain, some Z-rings identified as 'nascent' at time of treatment may have been misclassified 'mature' rings that had already condensed, and hence the % of Z-rings condensing post-PC19 in that strain may be artificially high.

We have added a note in the caption clarifying this.

19. Supplementary note 1: It would be helpful to provide a diagram if possible to help with the explanation.

We have now provided a diagram to Supp. Note 1 to better illustrate the derivation of the tilted circle model. We have also altered the notation in the derivation to match this schematic.

Additional author revisions:

We had neglected to mention the fluorophore we used to label PBP2B in the experiments shown in Fig. 3c-d. HaloTag-PBP2B was labelled with JF549 HaloTag ligand for all experiments using strain SH212. We have now mentioned this throughout the manuscript, and acknowledged the Lavis lab for generously donating these reagents.

In Fig. 4a, we had been displaying only 50 representative traces for each condition (untreated and PC19-treated) for clarity, but excluding the remaining traces from these plots was unnecessary. We have now included all traces as grey lines.

In Fig. 4d, we had excluded four data points that showed abnormally high treadmilling speeds ($>3\sigma$). However, this was unnecessary, and we have now included these four data points in the distributions.

We had misannotated the strain in Supp. Fig. 13 (revised: 20) as bAB217 (FtsA-mNeonGreen with FtsZ(D213A)). This has now been corrected as SH132 (mNeonGreen-PBP2B with FtsZ(D213A)). SH132 has been added to Supp. Table 1 (revised: 2), and description of its construction included in the "Strain Construction" part of Methods.

Editorial requests

POLICIES AND FORMS REQUIRED FOR RESUBMISSION

* Please complete or update the following checklist(s) to verify compliance with our research ethics and data reporting standards. Address all points on the checklist, revising your manuscript in response to the points if needed.

The form(s) must be downloaded and completed in Adobe Reader rather than opened in a web browser. Each form should be uploaded as a Related Manuscript file at the time of resubmission.

Editorial policy checklist:

<https://www.nature.com/documents/nr-editorial-policy-checklist.pdf>

Reporting summary:

Updated versions of these forms are included with the revised manuscript submission.

DATA AND CODE AVAILABILITY

* All Nature Communications manuscripts must include a “Data Availability” section after the Methods section but before the References. If any of the data can only be shared on request or are subject to restrictions, please specify the reasons and explain how, when, and by whom the data can be accessed. For more information on this policy and a list of examples, see:

<https://www.nature.com/documents/nr-data-availability-statements-data-citations.pdf>

See data and code availability statements L735-742 of the manuscript.

* We strongly encourage you to deposit all new data associated with the paper in a persistent repository where they can be freely and enduringly accessed. We recommend submitting the data to discipline-specific and community-recognized repositories; a list of repositories is provided here:

<http://www.nature.com/sdata/policies/repositories>

Source data for all figures presented in the paper and Supplementary Information, as well as raw movie files for all Supplementary Videos are available at

https://data.ncl.ac.uk/projects/FtsZ_treadmilling_is_essential_for_Z-ring_condensation_and_septal_constriction_initiation_in_Bacillus_subtilis_cell_division/92465 as noted L735-736.

* To maximise the reproducibility of research data, we strongly encourage you to provide a file containing the raw data underlying the following types of display items:

- Any reported means/averages in box plots, bar charts, and tables
- Dot plots/scatter plots, especially when there are overlapping points
- Line graphs

The data should be provided in a single Excel file with data for each figure/table in a separate sheet, or in multiple labelled files within a zipped folder. Name this file or folder ‘Source Data’, and include a brief description in your cover letter. The “Data Availability” section should also include the statement “Source data are provided with this paper.”

To learn more about our motivation behind this policy, please see:

<https://www.nature.com/articles/s41467-018-06012-8>

See above

* We also mandate the presentation of uncropped versions of any gels or blots, labelled with the relevant panel and identifying information such as the antibody used.

Raw uncropped western blot images are included in the manuscript source data files.

* Please replace your bar graphs with plots that feature information about the distribution of the underlying data. All data points should be shown for plots with a sample size less than 10. For larger sample sizes, please consider box-and-whisker or violin plots as alternatives. Measures of centrality, dispersion and/or error bars should be plotted and described in the figure legend.

Stacked bar charts are used only for categorical data in this manuscript (e.g. Figure 3b). Stacked bar charts are an appropriate format for this data type, as they represent all information about categorical data, i.e. number of data points per category.

Reviewer #1 (Remarks to the Author):

The authors of the manuscript "FtsZ treadmilling is essential for Z-ring condensation and septal constriction initiation in *Bacillus subtilis* cell division" have submitted a revised version of their work. They provide convincing answers to all questions raised in the first round and support their claims with several new experiments. Importantly, they clarify the use of fluorescent fusions and add more data on temperature sensitivity as well as new measurements with FtsZ mutants. I agree with the authors that these data provide good evidence that the effects they observe are not hampered by the use of fluorescent fusions or dominant negative effects. The reasons for immobile FtsZ filaments is not clear, yet, but the authors discuss this issue now and I think this issue is not easily answered/solved and may be left for future work.

I have just one question left:

On page 12 (line 330) the authors state that "observations show that cell growth rate is a major regulator of active septal constriction,..."

This is an important statement and should be discussed in the Discussion section and explained in more detail.

Altogether this is a very comprehensive and well conducted piece of work. Congratulations to the authors!

Reviewer #2 (Remarks to the Author):

I am satisfied with the Authors' responses to the reviewer comments and recommend publication of this manuscript.

Reviewer #4 (Remarks to the Author):

This paper by Whitley et al. contains important advancements in microscopy methods that will help advance the field of bacterial cell division and morphology. These methods are readily applicable to many questions, including the mechanisms of bacterial cell division, but one could imagine studying cell appendage dynamics, transport, elongosome, and so on.

While the paper includes some of the most accurate measurements yet presented for FtsZ and PG synthase movements as acquired by their novel VerCINI method, the novel biological insight is still somewhat limited. The authors demonstrate that in *B. subtilis*, the relationship between FtsZ dynamics and PG synthesis is similar to *S. aureus* in that once PG synthesis is initiated, FtsZ treadmilling appears to be non-essential, though it can still contribute to speed of constriction. This does rewrite the prior model in which *B. subtilis* PG synthesis was thought to be dependent on FtsZ treadmilling. Also, as the authors mention in their rebuttal, the VerCINI technique is responsible for the ability to measure how FtsZ treadmilling is a rate-determining factor in constriction. However, the assertion that FtsZ condensation is essential is not well-supported by the data, and this is a novel claim by this paper. The paper itself contradicts that conclusion with the data in Supplementary Figure 17.

The authors have done well at adding caveats where appropriate, addressing comments, and performing suggested experiments from each reviewer. Particularly, addressing my previous

comments that adding additional perturbations, such as a GTPase mutant and another chemical perturbation with a different mechanism, have more clearly illustrated how FtsZ dynamics are involved in regulation during the multiple steps of *B. subtilis* division. Also, I appreciate the author taking criticism with the extent to which their results may be extrapolated, such as delineating whether the results apply to *B. subtilis*, gram-positive bacteria, or the whole of the bacterial kingdom. I feel that the results of the simulations were not entirely convincing that there is not a loss of some filament tracking in dense rings and that the authors should include additional caveats of the practical limits of the data.

The paper really contributes a brilliant technique for studying divisome dynamics in greater detail and does establish that *B. subtilis* divisome dynamics have a level of FtsZ-independence, but the evidence about the roles of FtsZ condensation do not appear to support that condensation is essential for FtsZ function. In summary, I think that this paper is well written and includes some really in depth data that resolves well major aspects of FtsZ dynamics during *B. subtilis* division. Major comments:

Supplementary video 5/6 and supplementary figures 7 and 10: These videos of mature/early constricting and late constricting rings respectively look difficult for performing cluster tracking. While I believe the kymographs and data put forth in the rest of the paper, I do think that it is difficult to resolve whether clusters are lacking dynamics or that there are just multiple clusters in the plane of vision that are interfering with accurate measurements as the ring gets more constricted. The addition of simulations for the amount of filaments in a cell, with the corresponding kymographs, were helpful for demonstrating how many filaments would be necessary before tracking is unable to readily determine accurate filament movement. I do think it is important to list the constraints of the models used in these simulations, beyond those (photon noise, camera gain, read noise, etc) listed. I say this because if given an image of a mature-late constricting cell and these simulated traces without their estimated Z-ring densities and asked to place it to the closest related simulation (e.g. Figure 1gii), I would be most likely to say the simulation for 12x Z-ring density would most closely fit those of late-constricting rings. Therefore, I think these simulations may be missing assumptions for more accurate modeling and could be misrepresenting that each track may be accurately measured.

Line 127-133/figure 1H: Your interpretation that there is a significant change in speed between the nascent Z-ring clusters and the mature/early constricting appears to be a bit of an artifact, possibly of the automated method through which your tracking acquires speed or by the constraints of the microscopy. But a lot of the samples appear to be 'nearly immobile', which could mean that these were actually immobile clusters that were counted as slowly moving clusters. This would drastically decrease the apparent rate. Since that is a possibility, I would make a caveat that there could be limitations in the technique application that mistakes immobile as very slowly moving mobile clusters and this is why the average velocity is so low for the nascent FtsZ clusters. Alternatively, are there other automated approaches that could be applied specifically on slower molecules to show these are in fact moving very slowly as opposed to immobile?

Line 338: Based on the data in supplemental figure 17, FtsZ ring condensation is non-essential, at least for constriction initiation. Beyond constriction initiation, the condensed state is non-essential based on the proposed model that once the PG synthesis machinery is initiated, the process may occur independent of FtsZ treadmilling. Therefore, concluding that condensation of the Z-ring is essential is not supported by the data. Rather, as you have accurately included in your rebuttals, the condensation contributes to FtsZ-mediated rate improvement for constriction.

Minor comments:

Line 82: I would say "comparably wild-type" or "near wild-type" rather than "wild-type protein

levels” because there is no quantification like densitometry and this is a single example of protein levels in a single sample.

Line 84/Supplement Figure 2: I feel like this is slightly more than mild. Based on my understanding of the DABEST plots that you have in the supplemental figure, there appears to be a similar change in average between the 30C and 37C for SH130 with the distribution in the 37C sample showing a skew towards the WT length. Your conclusion in the paper though is that there is only mild elongation at 37C, when to me, it appears there is mild elongation at both 30C and 37C for the SH130 strain. Maybe an additional experiment showing that many have WT morphology, but a few have division defect, that lead to this elongation, would be a good follow-up here. Because the criticism here is not that the strain has a length defect, but rather that if you want to extrapolate WT dynamics, you must first want to demonstrate WT time to division or WT constriction dynamics. Essentially, the suggested experiment is to image cells using time-lapse microscopy to show how it is that some cells in a population filament. This could be especially beneficial considering many cells appear to be near the WT average for length. Also, across the data presented, is there a significant difference in length? It could be beneficial to include significance values on these types of graphs. Was width also measured for these cells?

Line 123/Supplementary figure 7: I don’t think that there is a consistency to the temporary arrest. Some appear to still have movement after apparent collision. In one case, there even appears to be the disappearance of a FtsZ cluster when a collision occurs and the second cluster appears to carry on directionally. Some do occasionally “arrest” or at least the apparent cluster movement appears to halt. I think an appropriate caveat to mention would be that not every single apparent collision results in temporary arrest.

Line 207: I thought a comment was made that no experiments relied on the results of rich media at 37C because of the mild filamentation that occurred in strain SH130. Is this a mistake or was this experiment performed at this temperature?

Line 291-293: Your presentation of the data is inconsistent here. You once give the IQR one way and in the next case present it in another format. Please try to be consistent with the data presentation for ease of reading and comparing different sets.

Line 320: Figure 4b displays constriction times better than 4a. I think this is the sub-figure you meant to cite.

Line 324: Figure 4d displays treadmilling speed, not 4b.

Figure 4/ FtsZ treadmilling accelerates constriction section: I think it would be beneficial to look at the percentage of “immobile” FtsZ clusters as had been performed for the data of Figure 1 for the different media and temperature conditions.

Supplementary figures (general comment): These DABEST plots are not particularly friendly for those unacquainted with them. I think a small explanation for how they function in more layman’s terms included in their section of the methods, as well as a re-direct to their section of the methods in the legend of Figure 4, where they are used in the main figures, would be beneficial to some readers.

MciZ experiment results: I think this could be an appropriate time to include the conclusion that this experiment was performed but due to the mentioned difficulties (time-scale too slow for appropriate experiments) that the results are not included.

Reviewer #1 (Remarks to the Author):

The authors of the manuscript “FtsZ treadmilling is essential for Z-ring condensation and septal constriction initiation in *Bacillus subtilis* cell division” have submitted a revised version of their work. They provide convincing answers to all questions raised in the first round and support their claims with several new experiments. Importantly, they clarify the use of fluorescent fusions and add more data on temperature sensitivity as well as new measurements with FtsZ mutants. I agree with the authors that these data provide good evidence that the effects they observe are not hampered by the use of fluorescent fusions or dominant negative effects. The reasons for immobile FtsZ filaments is not clear, yet, but the authors discuss this issue now and I think this issue is not easily answered/solved and may be left for future work.

We thank the reviewer for their kind words and interest in the manuscript. We respond to the reviewer’s final comment below.

I have just one question left:

On page 12 (line 330) the authors state that “observations show that cell growth rate is a major regulator of active septal constriction,...” This is an important statement and should be discussed in the Discussion section and explained in more detail.

We are glad that the reviewer finds this observation interesting – we left out more detailed discussion in the interests of brevity, but we are also fascinated by the remarkably high degree of coupling between constriction rate and overall cell growth rate, and aim to follow this up in the near future. We have now expanded this discussion as suggested (L408-412).

Altogether this is a very comprehensive and well conducted piece of work. Congratulations to the authors!

Reviewer #2 (Remarks to the Author):

I am satisfied with the Authors' responses to the reviewer comments and recommend publication of this manuscript.

We thank the reviewer for their recommendation and for careful reading of the manuscript.

Reviewer #4 (Remarks to the Author):

This paper by Whitley et al. contains important advancements in microscopy methods that will help advance the field of bacterial cell division and morphology. These methods are readily applicable to many questions, including the mechanisms of bacterial cell division, but one could imagine studying cell appendage dynamics, transport, elongasome, and so on.

While the paper includes some of the most accurate measurements yet presented for FtsZ and PG synthase movements as acquired by their novel VerCINI method, the novel biological insight is still somewhat limited. The authors demonstrate that in *B. subtilis*, the relationship between FtsZ dynamics and PG synthesis is similar to *S. aureus* in that once PG synthesis is initiated, FtsZ treadmilling appears to be non-essential, though it can still contribute to speed of constriction. This does rewrite the prior model in which *B. subtilis* PG synthesis was thought to be dependent on FtsZ treadmilling. Also, as the authors mention in their rebuttal, the VerCINI technique is responsible for the ability to measure how FtsZ treadmilling is a rate-determining factor in constriction. However, the assertion that FtsZ condensation is essential is not well-supported by the data, and this is a novel claim by this paper. The paper itself contradicts that conclusion with the data in Supplementary Figure 17.

We thank the reviewer for raising this point about the interpretation of the FtsZ(D213A) experiments. Although we believe that the data in Revision 1 are consistent with our conclusion that Z-ring condensation is essential for septal constriction, the reviewer rightly highlights that it would be useful to further investigate the intriguing intermediate Z-ring condensation phenotype induced by expression of FtsZ(D213A). We have investigated this phenotype in further detail and found that it adds useful additional insight into the mechanism of Z-ring condensation. Please see response to Major comment 3 for details.

The authors have done well at adding caveats where appropriate, addressing comments, and performing suggested experiments from each reviewer. Particularly, addressing my previous comments that adding additional perturbations, such as a GTPase mutant and another chemical perturbation with a different mechanism, have more clearly illustrated how FtsZ dynamics are involved in regulation during the multiple steps of *B. subtilis* division. Also, I appreciate the author taking criticism with the extent to which their results may be extrapolated, such as delineating whether the results apply to *B. subtilis*, gram-positive bacteria, or the whole of the bacterial kingdom. I feel that the results of the simulations were not entirely convincing that there is not a loss of some filament tracking in dense rings and that the authors should include additional caveats of the practical limits of the data.

We have performed additional analysis of the simulations to assess the possibility of bias in our analysis, discussed below in response to Major Comment 1. We find that measurements of the key analysis parameters of FtsZ filament speed and fraction of immobile FtsZ filaments are robust across a large range of filament densities and therefore conclude that we can have good confidence in the corresponding experimental measurements. We have expanded the discussion of possible limits and biases in the data based on the simulation analysis (see Major Comment 1).

The paper really contributes a brilliant technique for studying divisome dynamics in greater detail and does establish that *B. subtilis* divisome dynamics have a level of FtsZ-independence, but the

evidence about the roles of FtsZ condensation do not appear to support that condensation is essential for FtsZ function.

See above comment regarding Z-ring condensation and response to Major Comment 3.

In summary, I think that this paper is well written and includes some really in depth data that resolves well major aspects of FtsZ dynamics during *B. subtilis* division.

We thank the reviewer for their overall positive comments and constructive criticism. We believe we have now fully addressed all of their concerns.

Major comments:

Supplementary video 5/6 and supplementary figures 7 and 10: These videos of mature/early constricting and late constricting rings respectively look difficult for performing cluster tracking. While I believe the kymographs and data put forth in the rest of the paper, I do think that it is difficult to resolve whether clusters are lacking dynamics or that there are just multiple clusters in the plane of vision that are interfering with accurate measurements as the ring gets more constricted. The addition of simulations for the amount of filaments in a cell, with the corresponding kymographs, were helpful for demonstrating how many filaments would be necessary before tracking is unable to readily determine accurate filament movement. I do think it is important to list the constraints of the models used in these simulations, beyond those (photon noise, camera gain, read noise, etc) listed.

In section “Simulations of FtsZ filament dynamics” (L746-760) we include detailed description of simulation parameters including the simulation model for FtsZ filaments and Z-rings and the basis of Z-ring density estimation for the simulations. Simulation source code is also publically available. In the absence of requests for specific additional details, because the subsequent questions by the reviewer relate to FtsZ filament density we have focussed on adding further details about how the density is estimated, together with further analysis and discussion about the simulations extra discussion (Supplementary Note 4, Supplementary Figure 10)

I say this because if given an image of a mature-late constricting cell and these simulated traces without their estimated Z-ring densities and asked to place it to the closest related simulation (e.g. Figure 1gii), I would be most likely to say the simulation for 12x Z-ring density would most closely fit those of late-constricting rings. Therefore, I think these simulations may be missing assumptions for more accurate modeling and could be misrepresenting that each track may be accurately measured.

The reviewer rightly points out that visually, the filament density in kymographs of real mature/constricting Z-rings looks as if it is most comparable to a density somewhat in between the simulated 2x and 12x estimated mature density. Although we did not previously state this, the reason we performed the simulations over a large FtsZ filament density range was precisely to make sure that our measurements of filament dynamics were robust across a large measurement range, such that any inaccuracy in FtsZ filament density estimation would not be problematic. We now explain this rationale, the principles of the density estimates, and possible limitations of the density estimates in

Supplementary Note 4. We have also performed quantitative analysis of the bias introduced on all filament dynamics measurements (speed, processivity, lifetime and immobile filament fraction) in Supplementary Figure 10. These analyses show that although the fraction of simulated filaments detected drops as filament density increases, measurements of the key parameters of speed and especially immobile filament fraction remain remarkably robust across a large range of simulated densities likely to encompass the experimental conditions (Supplementary Note 4, Supplementary Figure 10Ci, D).

Interestingly, as we now note in Supplementary Note 4, we found that FtsZ filament lifetimes and processivity are underestimated at high FtsZ filament density. This does not affect any of the main conclusions of the manuscript, however based on this finding we have removed the comparison of FtsZ filament lifetimes across the different cell division stages (previously at L135).

Line 127-133/figure 1H: Your interpretation that there is a significant change in speed between the nascent Z-ring clusters and the mature/early constricting appears to be a bit of an artefact, possibly of the automated method through which your tracking acquires speed or by the constraints of the microscopy. But a lot of the samples appear to be 'nearly immobile', which could mean that these were actually immobile clusters that were counted as slowly moving clusters. This would drastically decrease the apparent rate. Since that is a possibility, I would make a caveat that there could be limitations in the technique application that mistakes immobile as very slowly moving mobile clusters and this is why the average velocity is so low for the nascent FtsZ clusters. Alternatively, are there other automated approaches that could be applied specifically on slower molecules to show these are in fact moving very slowly as opposed to immobile?

The reviewer correctly notes that filaments with low speeds (≤ 10 nm/s) in the violin plots of Fig. 1h could fairly be called 'nearly immobile' rather than 'immobile'. The exact nature of the immobile/nearly immobile filaments is not currently known. These could represent filaments that are truly immobile through interaction with partner proteins that are bound to the cell wall sacculus, or could represent nearly immobile filaments that are very slowly diffusing along the membrane. Regardless, it is clear from the distribution of speeds (Fig. 1h) that these represent a distinct dynamic population that is not consistent with treadmilling. Our conclusion that FtsZ filaments in nascent rings are overall less mobile than those of mature and constricting rings is not affected by a distinction of immobile vs. nearly immobile.

Regarding the method of analysis, we firstly note that we did not use an automated tracking method to sort samples into these two categories; we measured speeds directly from the kymographs (e.g. Fig. 1e-g) without any classification of 'mobile' or 'immobile', as described in the "FtsZ filament treadmilling analysis using VerCINI" section of Methods (L641-655). The resulting distribution of the measured speeds is Fig. 1h. We then used a simple threshold of 10 nm/s to separate 'mobile' and 'immobile'. Although simple, this threshold is only meant to quantify what is clear from the violin plots: there are two distinct populations with mean speeds of ~ 0 and ~ 25 nm/s, and the ~ 0 nm/s population is larger in nascent rings than in mature and constricting rings. Our conclusion that nascent Z-rings have a higher percentage of immobile filaments will therefore not change with an alternative analysis method.

Regarding the average speeds that we report for each state, these averages are for the entire distributions—‘mobile’ and ‘immobile’ together—and so do not depend on our threshold defining the two states. As correctly pointed out by the reviewer, the large percentage of immobile filaments in nascent Z-rings will therefore lower the average speed of FtsZ filaments for this division stage. We acknowledge this in the text (L119-122):

“In nascent Z-rings, we observed that FtsZ filaments are sparsely distributed, with **a mixture of treadmilling and immobile filaments** (Figure 1f, Supplementary Figure 9a, 35% immobile, filament speed < 10 nm/s), **leading to a low average filament speed** with large spread (median 19 nm/s, interquartile range, IQR, 5-31, N=526).”

In summary, our conclusion that the behaviour of FtsZ filaments in nascent rings is distinctly different from mature and constricting rings is not affected by our analysis method or definition of ‘immobile’.

Line 338: Based on the data in supplemental figure 17, FtsZ ring condensation is non-essential, at least for constriction initiation. Beyond constriction initiation, the condensed state is non-essential based on the proposed model that once the PG synthesis machinery is initiated, the process may occur independent of FtsZ treadmilling. Therefore, concluding that condensation of the Z-ring is essential is not supported by the data. Rather, as you have accurately included in your rebuttals, the condensation contributes to FtsZ-mediated rate improvement for constriction.

The reviewer raises a valid point here, that the FtsZ(D213A) data requires further analysis in order to fully understand how it fits with the other measurements. As correctly pointed out, the data we presented in Supp. Fig. 17 indicate that constriction can occur in the SH131 strain (expressing the mutant FtsZ(D213A)) even when the measured Z-ring thickness is large. Although we discussed this only very briefly in Revision 1, we interpreted this phenotype as “partial” Z-ring condensation caused by reduction but not total arrest of FtsZ treadmilling speed. However, an alternative possibility raised by the reviewer is that this result could be a case where Z-ring condensation is not required for constriction.

We therefore conducted additional measurements and analyses in order to discriminate between these possibilities.

We have added a new Supplementary Figure (Supp. Fig. 18) comparing the pre-constricted Z-ring states of wild-type and mutant. We have also revised Supp. Fig. 17 to avoid referring to mutant Z-rings as “did not condense” or “condensed.” We have now included the following further discussion on this topic in the Results section (L250-278), as well as a brief addition to the Discussion section (L382-385):

We investigated how FtsZ(D213A)-expressing cells could still constrict despite forming thick partially condensed Z-rings. First, we performed state detection on the D213A Z-rings to attempt to identify transitions between a nascent and condensed state. However, unlike in wild type cells, we could only reliably detect a single pre-constricted Z-ring state in the D213A cells (Supplementary Figure 18a-b). We thus compared the axial thickness of the D213A Z-

rings to that of WT nascent and mature Z-rings. We found that the thickness of the D213A Z-rings is intermediate between that of WT nascent and mature Z-rings. We also noticed that most pre-constricted D213A Z-rings looked quite dissimilar to the highly dynamic, non-condensed structures characteristic of the nascent division stage in wild-type cells, but more like stable, condensed mature Z-rings except with aberrant shapes (Supplementary Videos 1 and 17). This prompted us to measure the variation in Z-ring axial thickness over time for both wild type and D213A-expressing cells. We found that the variations in thickness for pre-constricted mutant Z-ring structures were small and comparable to those of mature wild-type Z-rings (Supplementary Figure 18d). On the other hand, the axial thickness of nascent Z-rings in wild-type cells was highly variable, consistent with their visibly dynamic appearance, and was considerably more variable than either wild-type mature Z-rings or FtsZ(D213A) pre-constricted Z-rings (Supplementary Figure 18c). Taken together these results show that in cells expressing GTPase-deficient FtsZ(D213A), FtsZ filaments rapidly aggregate into partially condensed Z-rings which are dense and structurally stable, but which have increased axial thickness compared to mature Z-rings in wild-type cells.

To further investigate the connection between condensation and septal constriction, we imaged wild-type and D213A-expressing cells with a membrane stain (Supplementary Figure 18e). In the mutant, we observed sharp membrane invaginations in many cases where aberrant Z-ring shapes occurred, indicating active but aberrant constriction. In contrast, we did not observe any clear cases of membrane invagination over areas corresponding to nascent Z-rings in wild-type cells. These observations indicate that the rapid treadmilling dynamics of FtsZ filaments in wild-type nascent Z-rings are likely required to form a compact Z-ring structure to mark the division septum to enable constriction. Reduced filament treadmilling and turnover in the FtsZ(D213A) mutant still allow assembly of a dense Z-ring structure, but locked into a thick, partially-condensed state that is less effective in directing cell division. This observation is reminiscent of how dynamic instability of microtubules allows them to rapidly conform to the complex intracellular geometry of eukaryotic cells.

In summary, these additional analyses strongly support the conclusion that Z-ring condensation is essential for septal constriction. However these data also provide further insight into Z-ring assembly, showing that Z-ring condensation state is not solely determined by overall Z-ring thickness, but also Z-ring structural stability over time.

Minor comments:

Line 82: I would say “comparably wild-type” or “near wild-type” rather than “wild-type protein levels” because there is no quantification like densitometry and this is a single example of protein levels in a single sample.

We have made this change. We note that Western blots experiments were conducted in duplicate and similar results were obtained each time (data not shown).

Line 84/Supplement Figure 2: I feel like this is slightly more than mild. Based on my understanding of the DABEST plots that you have in the supplemental figure, there appears to be a similar change in average between the 30C and 37C for SH130 with the distribution in the 37C sample showing a skew towards the WT length. Your conclusion in the paper though is that there is only mild elongation at 37C, when to me, it appears there is mild elongation at both 30C and 37C for the SH130 strain.

The reviewer raises a good point that the elongation phenotype appears to be similar at 30C and 37C for SH130. We have revised the statement in the main text accordingly (L83-84):

“Cell morphology analysis showed near native morphology with only a mild elongation phenotype at both 30°C and 37°C”

Maybe an additional experiment showing that many have WT morphology, but a few have division defect, that lead to this elongation, would be a good follow-up here. Because the criticism here is not that the strain has a length defect, but rather that if you want to extrapolate WT dynamics, you must first want to demonstrate WT time to division or WT constriction dynamics. Essentially, the suggested experiment is to image cells using time-lapse microscopy to show how it is that some cells in a population filament. This could be especially beneficial considering many cells appear to be near the WT average for length.

We think that the reviewer is suggesting that the longer average lengths we measure for the SH130 strain could result from two populations of cells: one with WT cell lengths and one with long, filamentous cells. If true, we would expect this to yield distributions of cell lengths with a significant population of outliers representing the filamentous cells. To see if our length distributions reflect this, we tried re-plotting these distributions while excluding data outside 3σ , 2σ , or 1σ from each mean (see figures below). Even with outliers removed, the difference in lengths between WT (PY79) and SH130 strains is robust. This shows that SH130 do not form a mixed population of short and filamentous cells. Rather, like WT cells, SH130 cells are non-filamentous, but are slightly longer on average than WT cells.

Regarding the reviewer's comment about extrapolating WT dynamics from the SH130 strain, we agree that use of SH130 strain to measure constriction rate could be potentially problematic due to the mild elongation phenotype. For this reason, we deliberately chose not to use the SH130 strain for any quantitative measurements of septal division rate (e.g. constriction time in Fig. 4). For these measurements we instead used the ME7 strain (natively expressing mNeonGreen-PBP2B), which was previously shown to have morphology similar to WT (Bisson-Filho et al. *Science* 2017 Fig. S3B). ME7 constriction time is also consistent with constriction measurements for a wide range of minimally perturbative septal-localizing fluorescent protein fusions (Bisson Filho et al (Erratum) *Science* 2020 Fig E1.F). ME7 constriction dynamics were not compared to WT, as constriction rate measurements of unlabelled wild type *B. subtilis* cells are not technically feasible.

Upon looking back at this we notice that we did not make explicit in the text that we used the ME7 strain for these measurements. **We have now updated the text to clarify this (L311).**

Also, across the data presented, is there a significant difference in length? It could be beneficial to include significance values on these types of graphs.

The magnitude and robustness of the difference in lengths is shown in Supp. Fig. 2 with the DABEST plots. We choose to measure effect size using DABEST plots rather than performing analysis of statistical significance due to the established problems with null hypothesis significance testing (eg Wasserstein & Lazar, *The American Statistician*, 2020). For further discussion on our choice of presenting effect size and DABEST plots over other methods, see our response to comment below, as well as new description in the Methods section of the main text.

Was width also measured for these cells?

We have now provided width measurements for these cells as Supp. Fig. 2b

Line 123/Supplementary figure 7: I don't think that there is a consistency to the temporary arrest. Some appear to still have movement after apparent collision. In one case, there even appears to be the disappearance of a FtsZ cluster when a collision occurs and the second cluster appears to carry on directionally. Some do occasionally "arrest" or at least the apparent cluster movement appears to halt.

I think an appropriate caveat to mention would be that not every single apparent collision results in temporary arrest.

We have now revised the text to make clear that apparent collision does not necessarily mean temporary arrest:

“Surprisingly, we frequently observed two or more treadmilling FtsZ filaments colliding, **often** leading to aggregation and temporary arrest of both filaments (Figure 1eii, Supplementary Figure 7).”

Line 207: I thought a comment was made that no experiments relied on the results of rich media at 37C because of the mild filamentation that occurred in strain SH130. Is this a mistake or was this experiment performed at this temperature?

This is not a mistake – this experiment was performed at 37C. However, the same experiment was also performed at 30C (see L212-213 and Supp. Fig. 14) to ensure that the results were not affected by the higher temperature. As the results are very similar at both temperatures, we have chosen to present the 37C data in the main text (Fig. 3a-b) to be more comparable to the data presented in Fig. 3c-d (which is also at 37C).

Line 291-293: Your presentation of the data is inconsistent here. You once give the IQR one way and in the next case present it in another format. Please try to be consistent with the data presentation for ease of reading and comparing different sets.

Interquartile range and confidence interval are used for different purposes here. IQR is used in the first case to describe the spread of the data (i.e. how much the constriction times vary). Use of 95% CI (or S.E.M.) would not be appropriate, since we are not describing how accurately we know the ‘true’ constriction time, but the variation in our measured data. 95% CI is used in the second case to describe how accurately the increase in constriction time is known between conditions. Use of IQR (or S.D.) would not be appropriate for that purpose. Throughout the manuscript we have clearly indicated which measure of spread or uncertainty we have used.

Line 320: Figure 4b displays constriction times better than 4a. I think this is the sub-figure you meant to cite.

We have made this change.

Line 324: Figure 4d displays treadmilling speed, not 4b.

We thank the reviewer for pointing this out. We have made this change.

Figure 4/ FtsZ treadmilling accelerates constriction section: I think it would be beneficial to look at the percentage of “immobile” FtsZ clusters as had been performed for the data of Figure 1 for the different media and temperature conditions.

We agree with the reviewer that it would be interesting to see the percentage of immobile FtsZ filaments across different media and temperature conditions. However, we do not present these results for a few reasons.

Firstly, we decided not to measure FtsZ treadmill dynamics across different temperatures using the (temperature-sensitive) SH130 strain, which has fully labelled FtsZ. Instead, we used a separate (non-temperature-sensitive) strain expressing a dilute exogenous label of FtsZ (strain bWM4). Although this enabled us to measure treadmill speeds across multiple conditions (Fig. 4d), the percentage of mobile/immobile filaments may be affected by the heterogeneous population of FtsZ (labelled and unlabelled) and the levels of FtsZ in the cell (since labelled FtsZ is expressed from an inducible promoter). So, it may be unwise to draw conclusions from this.

Secondly, we performed this experiment using TIRF illumination on horizontal cells rather than VerCINI since we were only interested in measuring treadmill speed (see Supp. Fig. 22 (23 in this revision)). Since we can only observe the bottom 100-200 nm of the Z-ring with TIRF illumination, we cannot observe the entire ring, and so we cannot obtain percentage mobile/immobile filaments from the data (even if the strain were conducive to such a measurement).

However, we notice that we did not mention in the text that the results shown in Fig. 4d were obtained from TIRF measurements. We also notice that we neglected to reference Supp. Fig. 22 (23 in this revision) in this section, which shows examples of these measurements. **We have now explicitly stated that the measurements were done using TIRF illumination (1353) and referenced Supp. Fig. 23.**

Supplementary figures (general comment): These DABEST plots are not particularly friendly for those unacquainted with them. I think a small explanation for how they function in more layman's terms included in their section of the methods, as well as a re-direct to their section of the methods in the legend of Figure 4, where they are used in the main figures, would be beneficial to some readers.

We have now expanded our explanation of the concept of DABEST and estimation statistics with the following additional discussion (1728-736):

“The basic principle of a DABEST plot is to calculate and plot the average (here median) difference between two conditions, together with the 95% confidence interval on that difference. This has the advantage over null hypothesis significance testing (NHST) of visualising not just the probable reproducibility of a difference between two conditions, but also the estimated magnitude of the difference between the two conditions. This is useful because it allows assessment of whether magnitude of difference is physically/ biologically meaningful, rather than just reproducible. For comparison to NHST analysis, we note that a median difference DABEST plot where the 95 % CIs do not overlap the 0-difference axis would indicate rejection of the null hypothesis that the two population medians are equal at $p=0.05$ significance level.”

MciZ experiment results: I think this could be an appropriate time to include the conclusion that this experiment was performed but due to the mentioned difficulties (time-scale too slow for appropriate experiments) that the results are not included.

We have now included mention of the results from MciZ in the main text (L230-233).

Reviewer #4 (Remarks to the Author):

Whitley et al have accepted all the criticisms and performed the necessary experiments to demonstrate their claims beyond a reasonable doubt. Particularly, I think that the experiments done in the D213A background provide clearer evidence that condensation is important for constriction events. The results of these experiment do appear to show that FtsZ still condenses before constriction even in a background where FtsZ is less likely to condense. I believe this manuscript could be accepted in its current form with just a few of the suggested minor comments.

Minor comments:

L232 – This line is missing the word “to” (too slow to draw conclusions).

L256-259 – This sentence is fairly hard to read on a first pass. If possible, a better wording should be used.

D213A experiments: While the experiment in a D213A background does show that condensation precedes invagination events, comparing to WT is not the best comparison for these experiments. A better strain comparison would be one in which FtsZ-GFP is expressed natively and FtsZ-WT is expressed ectopically using 10uM IPTG just as was done for D213A. However, chances are exceedingly low that this will change the interpreted conclusions, so I do not think it is absolutely necessary to re-perform experiment in this strain before publication, though it would be the better comparison.

Supplement figure 2/minor comment from last round of comments:

I think my explanation of concern was not very clear. I was suggesting that to get an overall population average to be greater than WT, due to cell filamentation there are two possibilities as I see it: 1) the cells must either all increase in time required to constrict, or 2) a sub-population of cells fail to divide for an extended period of time and then eventually do divide. The distribution of the population suggests that the latter is the case for these cells, which suggests that overwhelmingly the majority of cells have close to WT dynamics. However, to be certain that the constriction rate is close to that of WT, time-lapse and constriction rate analysis would be best. This would increase the ability to draw WT conclusions from measurements in the SH130 strain. But this is not necessary for the manuscript to be accepted, in my opinion.

Reviewer #4 (Remarks to the Author):

Whitley et al have accepted all the criticisms and performed the necessary experiments to demonstrate their claims beyond a reasonable doubt. Particularly, I think that the experiments done in the D213A background provide clearer evidence that condensation is important for constriction events. The results of these experiment do appear to show that FtsZ still condenses before constriction even in a background where FtsZ is less likely to condense. I believe this manuscript could be accepted in its current form with just a few of the suggested minor comments.

We thank the reviewer for their time reviewing the manuscript and their helpful suggestions. We respond to their remaining comments below.

Minor comments:

L232 – This line is missing the word “to” (too slow to draw conclusions).

We have made this change.

L256-259 – This sentence is fairly hard to read on a first pass. If possible, a better wording should be used.

We have revised this sentence to make it more clear:

“We also noticed that most pre-constricted D213A Z-rings looked quite dissimilar to the highly dynamic nascent Z-rings seen in wild-type cells, but instead appeared similar to the less dynamic mature Z-rings except with aberrant shapes (Supplementary Videos 1 and 17)”

D213A experiments: While the experiment in a D213A background does show that condensation precedes invagination events, comparing to WT is not the best comparison for these experiments. A better strain comparison would be one in which FtsZ-GFP is expressed natively and FtsZ-WT is expressed ectopically using 10uM IPTG just as was done for D213A. However, chances are exceedingly low that this will change the interpreted conclusions, so I do not think it is absolutely necessary to re-perform experiment in this strain before publication, though it would be the better comparison.

While we agree that this suggestion is interesting, as noted by the reviewer it is unlikely to affect the conclusions of the manuscript so we have not performed this experiment.

Supplement figure 2/minor comment from last round of comments:

I think my explanation of concern was not very clear. I was suggesting that to get an overall population average to be greater than WT, due to cell filamentation there are two possibilities as I see it: 1) the cells must either all increase in time required to constrict, or 2) a sub-population of cells fail to divide for an extended period of time and then eventually do divide. The distribution of the population suggests that the latter is the case for these cells, which suggests that overwhelmingly the majority of cells have close to WT dynamics. However, to be certain that the constriction rate is close to that of WT, time-lapse and constriction rate analysis would be best. This would increase the ability to draw

WT conclusions from measurements in the SH130 strain. But this is not necessary for the manuscript to be accepted, in my opinion.

While we agree that this suggestion is interesting, as noted by the reviewer it is unlikely to affect the conclusions of the manuscript so we have not performed this experiment.